# LEARNING TO LINK

**Maria-Florina Balcan**
Carnegie Mellon University
ninamf@cs.cmu.edu

**Travis Dick**
University of Pennsylvania
tbd@seas.upenn.edu

**Manuel Lang**
Karlsruhe Institute of Technology
manuel.lang@student.kit.edu

## ABSTRACT

Clustering is an important part of many modern data analysis pipelines, including network analysis and data retrieval. There are many different clustering algorithms developed by various communities, and it is often not clear which algorithm will give the best performance on a specific clustering task. Similarly, we often have multiple ways to measure distances between data points, and the best clustering performance might require a non-trivial combination of those metrics. In this work, we study data-driven algorithm selection and metric learning for clustering problems, where the goal is to simultaneously learn the best algorithm and metric for a specific application. The family of clustering algorithms we consider is parameterized linkage based procedures that includes single and complete linkage. The family of distance functions we learn over are convex combinations of base distance functions. We design efficient learning algorithms which receive samples from an application-specific distribution over clustering instances and learn a near-optimal distance and clustering algorithm from these classes. We also carry out a comprehensive empirical evaluation of our techniques showing that they can lead to significantly improved clustering performance on real-world datasets.

## 1 INTRODUCTION

**Overview.** Clustering is an important component of modern data analysis. For example, we might cluster emails as a pre-processing step for spam detection, or we might cluster individuals in a social network in order to suggest new connections. There are a myriad of different clustering algorithms, and it is not always clear what algorithm will give the best performance on a specific clustering task. Similarly, we often have multiple different ways to measure distances between data points, and it is not obvious which distance metric will lead to the best performance. In this work, we study data-driven algorithm selection and metric learning for clustering problems, where the goal is to use data to simultaneously learn the best algorithm and metric for a specific application such as clustering emails or users of a social network. An application is modeled as a distribution over clustering tasks, we observe an i.i.d. sample of clustering instances drawn from that distribution, and our goal is to choose an approximately optimal algorithm from a parameterized family of algorithms (according to some well-defined loss function). This corresponds to settings where we repeatedly solve clustering instances (e.g., clustering the emails that arrive each day) and we want to use historic instances to learn the best clustering algorithm and metric.

The family of clustering algorithms we learn over consists of parameterized linkage based procedures and includes single and complete linkage, which are widely used in practice and optimal in many cases (Awasthi et al., 2014; Saeed et al., 2003; White et al., 2010; Awasthi et al., 2012; Balcan and Liang, 2016; Grosswendt and Roeglin, 2015). The family of distance metrics we learn over consists of convex combinations of base distance functions. We design efficient learning algorithms that receive samples from an application-specific distribution over clustering instances and simultaneously learn both a near-optimal distance metric and clustering algorithm from these classes. We contribute to a recent line of work that provides learning-theoretic guarantees for data-driven algorithm configuration (Gupta and Roughgarden, 2017; Balcan et al., 2017; 2018a;b; 2019). These papers analyze the intrinsic complexity of parameterized algorithm families in order to provide sample complexity guarantees, that is, bounds on the number of sample instances needed in order to find an approximately optimal algorithm for a given application domain. Our results build on the work of Balcan et al. (2017), who studied the problem of learning the best clustering algorithm from a class of linkage based procedures, but did not study learning the best metric. In addition to our sample complexity guarantees, we develop a number of algorithmic tools that enable learning application specific clustering algorithms and metrics for realistically large clustering instances. We use our efficient implementations to conduct comprehensive experiments on clustering domains derived from both real-world and synthetic datasets. These experiments demonstrate that learning

application-specific algorithms and metrics can lead to significant performance improvements over standard algorithms and metrics.

**Our Results.** We study linkage-based clustering algorithms that take as input a clustering instance $S$ and output a hierarchical clustering of $S$ represented as a binary *cluster tree*. Each node in the tree represents a cluster in the data at one level of granularity, with the leaves corresponding to individual data points and the root node corresponding to the entire dataset. Each internal node represents a cluster obtained by merging its two children. Linkage-based clustering algorithms build a cluster tree from the leaves up, starting with each point belonging to its own cluster and repeatedly merging the "closest" pair of clusters until only one remains. The parameters of our algorithm family control both the metric used to measure pointwise distances, as well as how the linkage algorithm measures distances between clusters (in terms of the distances between their points).

This work has two major contributions. Our first contribution is to provide sample complexity guarantees for learning effective application-specific distance metrics for use with linkage-based clustering algorithms. The key challenge is that, if we fix a clustering algorithm from the family we study and a single clustering instance $S$, the algorithm output is a piecewise constant function of our metric family's parameters. This implies that, unlike many standard learning problems, the loss we want to minimize is very sensitive to the metric parameters and small perturbations to the optimal parameters can lead to high loss. Our main technical insight is that for any clustering instance $S$, we can partition the parameter space of our metric family into a relatively small number of regions such that the ordering over pairs of points in $S$ given by the metric is constant on each region. The clustering output by all algorithms in the family we study only depends on the ordering over pairs of points induced by the metric, and therefore their output is also a piecewise constant function of the metric parameters with not too many pieces. We leverage this structure to bound the intrinsic complexity of the learning problem, leading to uniform convergence guarantees. By combining our results with those of Balcan et al. (2017), we show how to simultaneously learn both an application-specific metric and linkage algorithm.

Our second main contribution is a comprehensive empirical evaluation of our proposed methods, enabled by new algorithmic insights for efficiently learning application-specific algorithms and metrics from sample clustering instances. For any fixed clustering instance, we show that we can use an *execution tree* data structure to efficiently construct a coarse partition of the joint parameter space so that on each region the output clustering is constant. Roughly speaking, the execution tree compactly describes all possible sequences of merges the linkage algorithm might make together with the parameter settings for the algorithm and metric that lead to that merge sequence. The learning procedure proposed by Balcan et al. (2017) takes a more combinatorial approach, resulting in partitions of the parameter space that have many unnecessary regions and increased overall running time. Balcan et al. (2018a) and Balcan et al. (2018b) also use an execution tree approach for different algorithm families, however their specific approaches to enumerating the tree are not efficient enough to be used in our setting. We show that using a depth-first traversal of the execution tree leads to significantly reduced memory requirements, since in our setting the execution tree is shallow but very wide.

Using our efficient implementations, we evaluate our learning algorithms on several real world and synthetic clustering applications. We learn the best algorithm and metric for clustering applications derived from MNIST, CIFAR-10, Omniglot, Places2, and a synthetic rings and disks distribution. Across these different tasks the optimal clustering algorithm and metric vary greatly. Moreover, in most cases we achieve significant improvements in clustering quality over standard clustering algorithms and metrics.

**Related work.** Gupta and Roughgarden (2017) introduced the theoretical framework for analyzing algorithm configuration problems that we study in this work. They provide sample complexity guarantees for greedy algorithms for canonical subset selection problems including the knapsack problem, maximum weight independent set, and machine scheduling.

Some recent works provide sample complexity guarantees for learning application-specific clustering algorithms. Balcan et al. (2017) consider several parameterized families of linkage based clustering algorithms, one of which is a special case of the family studied in this paper. Their sample complexity results are also based on showing that for a single clustering instance, we can find a partitioning of the algorithm parameter space into regions where the output clustering is constant. The families of linkage procedures they study have a single parameter, while our linkage algorithm and metric families have multiple. Moreover, they suppose we are given a fixed metric for each clustering instance and do not study the problem of learning an application-specific metric. Balcan et al. (2018b) study the related problem of learning the best initialization procedure and local search method to use in a clustering algorithm inspired by Lloyd's method for $k$-means clustering. Their sample complexity results are again based on demonstrating that for any clustering instance, there exists a partitioning of the parameter space on which the algorithm's output is constant.

The parameter space partitions in both of these related works are defined by linear separators. Due to the interactions between the distance metric and the linkage algorithm, our partitions are defined by quadratic functions.

The procedures proposed by prior work for finding an empirically optimal algorithm for a collection of problem instances roughly fall into two categories: combinatorial approaches and approaches based on an execution-tree data structure. Gupta and Roughgarden (2017) and Balcan et al. (2017) are two examples of the combinatorial approach. They show that the boundaries in the constant-output partition of the algorithm parameter space always occur at the solutions to finitely many equations that depend on the problem instance. To find an empirically optimal algorithm, they find all solutions to these problem-dependent equations to explicitly construct a partition of the parameter space. Unfortunately, only a small subset of the solutions are actual boundaries in the partition. Consequently, their partitions contain many extra regions and suffer from long running times. The execution-tree based approaches find the coarsest possible partitioning of the parameter space such that the algorithm output is constant. Balcan et al. (2018b) and Balcan et al. (2018a) both use execution trees to find empirically optimal algorithm parameters for different algorithm families. However, the specific algorithms used to construct and enumerate the execution tree are different from those explored in this paper and are not suitable in our setting.

## 2    LEARNING CLUSTERING ALGORITHMS

The problem we study is as follows. Let $\mathcal{X}$ be a data domain. A clustering instance consists of a point set $S = \{x_1, \ldots, x_n\} \subset \mathcal{X}$ and an (unknown) target clustering $\mathcal{Y} = (C_1, \ldots, C_k)$, where the sets $C_1, \ldots, C_k$ partition $S$ into $k$ clusters. Linkage-based clustering algorithms output a hierarchical clustering of the input data, represented by a cluster tree. We measure the agreement of a cluster tree $T$ with the target clustering $\mathcal{Y} = (C_1, \ldots, C_k)$ in terms of the Hamming distance between $\mathcal{Y}$ and the closest pruning of $T$ into $k$ clusters (i.e., $k$ disjoint subtrees that contain all the leaves of $T$). More formally, we define the loss $\ell(T, \mathcal{Y}) = \min_{P_1, \ldots, P_k} \min_{\sigma \in \mathbb{S}_n} \frac{1}{|S|} \sum_{i=1}^{k} |C_i \setminus P_{\sigma_i}|$, where $A \setminus B$ denotes set difference, the first minimum is over all prunings $P_1, \ldots, P_k$ of the cluster tree $T$, and the second minimum is over all permutations of the $k$ cluster indices. This formulation allows us to handle the case where each clustering task has a *different number of clusters*, and where the desired number might not be known in advance. Our analysis applies to any loss function $\ell$ measuring the quality of the output cluster tree $T$, but we focus on the Hamming distance for simplicity. Given a distribution $\mathcal{D}$ over clustering instances (i.e., point sets together with target clusterings), our goal is to find the algorithm $A$ from a family $\mathcal{A}$ with the lowest expected loss for an instance sampled from $\mathcal{D}$. As training data, we assume that we are given an i.i.d. sample of clustering instances annotated with their target clusterings drawn from the application distribution $\mathcal{D}$.

We study linkage-based clustering algorithms. These algorithms construct a hierarchical clustering of a point set by starting with each point belonging to a cluster of its own and then they repeatedly merge the closest pair of clusters until only one remains. There are two distinct notions of distance at play in linkage-based algorithms: first, the notion of distance between pairs of points (e.g., Euclidean distance between feature vectors, edit distance between strings, or the Jaccard distance between sets). Second, these algorithms must define a distance function between clusters, which we refer to as a merge function to avoid confusion. A merge function $D$ defines the distance between a pair of clusters $A, B \subset \mathcal{X}$ in terms of the pairwise distances given by a metric $d$ between their points. For example, single linkage uses the merge function $D_{\min}(A, B; d) = \min_{a \in A, b \in B} d(a, b)$ and complete linkage uses the merge function $D_{\max}(A, B; d) = \max_{a \in A, b \in B} d(a, b)$.

Our parameterized family of linkage-based clustering algorithms allows us to vary both the metric used to measure distances between points, as well as the merge function used to measure distances between clusters. To vary the metric, we suppose we have access to $L$ metrics $d_1, \ldots, d_L$ defined on our data universe $\mathcal{X}$, and our goal is to find the best convex combination of those metrics. That is, for any parameter vector $\boldsymbol{\beta} \in \Delta_L = \{\boldsymbol{\beta} \in [0, 1]^L \mid \sum_i \beta_i = 1\}$, we define a metric $d_{\boldsymbol{\beta}}(x, x') = \sum_i \beta_i \cdot d_i(x, x')$. This definition is suitable across a wide range of applications, since it allows us to learn the best combination of a given set of metrics for the application at hand. Similarly, for varying the merge function, we suppose we have $L'$ merge functions $D_1, \ldots, D_{L'}$. For any parameter $\boldsymbol{\alpha} \in \Delta_{L'}$, define the merge function $D_{\boldsymbol{\alpha}}(A, B; d) = \sum_i \alpha_i D_i(A, B; d)$. For each pair of parameters $\boldsymbol{\beta} \in \Delta_L$ and $\boldsymbol{\alpha} \in \Delta_{L'}$, we obtain a different clustering algorithm (i.e., one that repeatedly merges the pair of clusters minimizing $D_{\boldsymbol{\alpha}}(\cdot, \cdot; d_{\boldsymbol{\beta}})$). Pseudocode for this method is given in Algorithm 1. In the pseudocode, clusters are represented by binary trees with leaves corresponding to the points belonging to that cluster. For any clustering instance $S \subset \mathcal{X}$, we let $\mathcal{A}_{\boldsymbol{\alpha}, \boldsymbol{\beta}}(S)$ denote the cluster tree output by Algorithm 1 when run with parameter vectors $\boldsymbol{\alpha}$ and $\boldsymbol{\beta}$.

---

**Algorithm 1** Linkage Clustering

**Input:** Metrics $d_1, \ldots, d_L$, merge functions $D_1, \ldots, D_{L'}$, points $x_1, \ldots, x_n \in \mathcal{X}$, parameters $\boldsymbol{\alpha}$ and $\boldsymbol{\beta}$.
1. Let $\mathcal{N} = \{\text{Leaf}(x_1), \ldots, \text{Leaf}(x_n)\}$ be the initial set of nodes (one leaf per point).
2. While $|\mathcal{N}| > 1$
    (a) Let $A, B \in \mathcal{N}$ be the clusters in $\mathcal{N}$ minimizing $D_{\boldsymbol{\alpha}}(A, B; d_{\boldsymbol{\beta}})$.
    (b) Remove clusters $A$ and $B$ from $\mathcal{N}$ and add $\text{Node}(A, B)$ to $\mathcal{N}$.
3. Return the cluster tree (the only element of $\mathcal{N}$).

---

First, we provide sample complexity results that hold for any collection of metrics $d_1, \ldots, d_L$ and any collection of merge functions $D_1, \ldots, D_{L'}$ that belong to the following family:

**Definition 1.** A merge function $D$ is *2-point-based* if for any pair of clusters $A, B \subset \mathcal{X}$ and any metric $d$, there exists a pair of points $(a, b) \in A \times B$ such that $D(A, B; d) = d(a, b)$. Moreover, the pair of points defining the merge distance must depend only on the ordering of pairwise distances. More formally, if $d$ and $d'$ are two metrics s.t. for all $a, a' \in A$ and $b, b' \in B$, we have $d(a, b) \leq d(a', b')$ if and only if $d'(a, b) \leq d'(a', b')$, then $D(A, B; d) = d(a, b)$ implies that $D(A, B; d') = d'(a, b)$.

For example, both single and complete linkage are 2-point-based merge functions, since they output the distance between the closest or farthest pair of points, respectively.

**Theorem 1.** *Fix any metrics $d_1, \ldots, d_L$, 2-point-based merge functions $D_1, \ldots, D_{L'}$, and distribution $\mathcal{D}$ over clustering instances with at most $n$ points. For any parameters $\epsilon > 0$ and $\delta > 0$, let $(S_1, \mathcal{Y}_1), \ldots, (S_N, \mathcal{Y}_N)$ be an i.i.d. sample of $N = O\left(\frac{1}{\epsilon^2}\left((L' + L)^2 L' \log\left(\frac{(L'+L)^2 L' n}{\epsilon^2}\right) + \log\left(\frac{1}{\delta}\right)\right)\right) = \tilde{O}\left(\frac{(L'+L)^2 L'}{\epsilon^2}\right)$ clustering instances with target clusterings drawn from $\mathcal{D}$. Then with probability at least $1 - \delta$ over the draw of the sample, we have*

$$\sup_{(\boldsymbol{\alpha}, \boldsymbol{\beta}) \in \Delta_{L'} \times \Delta_L} \left| \frac{1}{N} \sum_{i=1}^{N} \ell(\mathcal{A}_{\boldsymbol{\alpha}, \boldsymbol{\beta}}(S_i), \mathcal{Y}_i) - \mathbb{E}_{(S, \mathcal{Y}) \sim \mathcal{D}}\left[\ell(\mathcal{A}_{\boldsymbol{\alpha}, \boldsymbol{\beta}}(S), \mathcal{Y})\right] \right| \leq \epsilon.$$

The key step in the proof of Theorem 1 is to show that for any clustering instance $S$ with target clustering $\mathcal{Y}$, the function $(\boldsymbol{\alpha}, \boldsymbol{\beta}) \mapsto \ell(\mathcal{A}_{\boldsymbol{\alpha}, \boldsymbol{\beta}}(S), \mathcal{Y})$ is piecewise constant with not too many pieces and where each piece is simple. Intuitively, this guarantees that for any collection of clustering instances, we cannot see too much variation in the loss of the algorithm on those instances as we vary over the parameter space. In Appendix A we use this fact to bound the empirical Rademacher complexity of the learning problem and obtain the uniform convergence guarantee in Theorem 1. In the remainder of this section, we prove the key structural property.

We let $\boldsymbol{\zeta} = (\boldsymbol{\alpha}, \boldsymbol{\beta}) \in \Delta_{L'} \times \Delta_L$ denote a pair of parameter vectors for Algorithm 1, viewed as a vector in $\mathbb{R}^{L'+L}$. Our parameter space partition will be induced by the sign-pattern of $M$ quadratic functions.

**Definition 2** (Sign-pattern Partition). The sign-pattern partition of $\mathbb{R}^p$ induced $M$ functions $f_1, \ldots, f_M : \mathbb{R}^p \to \mathbb{R}$ is defined as follows: Let $F : \mathbb{R}^p \to \{\pm 1\}^M$ be the function $F(\boldsymbol{\zeta}) = \left(\text{sign}(f_1(\boldsymbol{\zeta})), \ldots, \text{sign}(f_M(\boldsymbol{\zeta}))\right)$. Two points $\boldsymbol{\zeta}, \boldsymbol{\zeta}' \in \mathbb{R}^p$ belong to the same region in the partition iff $F(\boldsymbol{\zeta}) = F(\boldsymbol{\zeta}')$. Each region is of the form $\mathcal{Z} = \{\boldsymbol{\zeta} \in \mathbb{R}^p | F(\boldsymbol{\zeta}) = \boldsymbol{b}\}$, for some sign-pattern vector $\boldsymbol{b} \in \{\pm 1\}^M$.

We show that for any fixed metrics $d_1, \ldots, d_L$ and clustering instance $S = \{x_1, \ldots, x_n\} \subset \mathcal{X}$, we can find a sign-pattern partitioning of $\Delta_L$ induced by linear functions such that, on each region, the ordering over pairs of points in $S$ induced by the metric $d_{\boldsymbol{\beta}}$ is constant. An important consequence of this result is that for each region $\mathcal{Z}$ in this partitioning of $\Delta_L$, the following holds: For any 2-point-based merge function $D$ and any pair of clusters $A, B \subset S$, there exists a pair of points $(a, b) \in A \times B$ such that $D(A, B; d_{\boldsymbol{\beta}}) = d_{\boldsymbol{\beta}}(a, b)$ for all $\boldsymbol{\beta} \in \mathcal{Z}$. In other words, restricted to $\boldsymbol{\beta}$ parameters belonging to $\mathcal{Z}$, the same pair of points $(a, b)$ defines the $D$-merge distance for the clusters $A$ and $B$.

**Lemma 1.** *Fix any metrics $d_1, \ldots, d_L$ and a clustering instance $S \subset \mathcal{X}$. There exists a set $\mathcal{H}$ of $O(|S|^4)$ linear functions mapping $\mathbb{R}^L$ to $\mathbb{R}$ with the following property: if two metric parameters $\boldsymbol{\beta}, \boldsymbol{\beta}' \in \Delta_L$ belong to the same region in the sign-pattern partition induced by $\mathcal{H}$, then the ordering over pairs of points in $S$ given by $d_{\boldsymbol{\beta}}$ and $d_{\boldsymbol{\beta}'}$ are the same. That is, for all points $a, b, a', b' \in S$ we have $d_{\boldsymbol{\beta}}(a, b) \leq d_{\boldsymbol{\beta}}(a', b')$ iff $d_{\boldsymbol{\beta}'}(a, b) \leq d_{\boldsymbol{\beta}'}(a', b')$.*

*Proof sketch.* For any pair of points $a, b \in S$, the distance $d_{\boldsymbol{\beta}}(a, b)$ is a linear function of the parameter $\boldsymbol{\beta}$. Therefore, for any four points $a, b, a', b' \in S$, we have that $d_{\boldsymbol{\beta}}(a, b) \leq d_{\boldsymbol{\beta}}(a', b')$ iff $h_{a,b,a',b'}(\boldsymbol{\beta}) \leq 0$, where $h_{a,b,a',b'}$ is the linear

function given by $h_{a,b,a',b'}(\boldsymbol{\beta}) = d_{\boldsymbol{\beta}}(a,b) - d_{\boldsymbol{\beta}}(a',b')$. Let $\mathcal{H} = \{h_{a,b,a',b'} \mid a,b,a',b' \in S\}$ be the collection of all such linear functions arising from any subset of 4 points. On each region of the sign-pattern partition induced by $\mathcal{H}$, all comparisons of pairwise distances in $S$ are fixed, implying that the ordering over pairs of points in $S$ is fixed. $\qquad\square$

Building on Lemma 1, we now prove the main structural property of Algorithm 1. We argue that for any clustering instance $S \subset \mathcal{X}$, there is a partition induced by quadratic functions of $\Delta_{L'} \times \Delta_L \subset \mathbb{R}^{L'+L}$ over $\boldsymbol{\alpha}$ and $\boldsymbol{\beta}$ into regions such that on each region, the ordering over all pairs of clusters according to the merge distance $D_{\boldsymbol{\alpha}}(\cdot,\cdot;d_{\boldsymbol{\beta}})$ is fixed. This implies that for all $(\boldsymbol{\alpha},\boldsymbol{\beta})$ in one region of the partition, the output of Algorithm 1 when run on $S$ is constant, since the algorithm output only depends on the ordering over pairs of clusters in $S$ given by $D_{\boldsymbol{\alpha}}(\cdot,\cdot;d_{\boldsymbol{\beta}})$.

**Lemma 2.** *Fix any metrics $d_1,\ldots,d_L$, any 2-point-based merge functions $D_1,\ldots,D_{L'}$, and clustering instance $S \subset \mathcal{X}$. There exists a set $\mathcal{Q}$ of $O(|S|^{4L'})$ quadratic functions defined on $\mathbb{R}^{L'+L}$ so that if parameters $(\boldsymbol{\alpha},\boldsymbol{\beta})$ and $(\boldsymbol{\alpha}',\boldsymbol{\beta}')$ belong to the same region of the sign-pattern partition induced by $\mathcal{Q}$, then the ordering over pairs of clusters in $S$ given by $D_{\boldsymbol{\alpha}}(\cdot,\cdot;d_{\boldsymbol{\beta}})$ and $D_{\boldsymbol{\alpha}'}(\cdot,\cdot;d_{\boldsymbol{\beta}'})$ is the same. That is, for all clusters $A, B, A', B' \subset S$, we have that $D_{\boldsymbol{\alpha}}(A,B;d_{\boldsymbol{\beta}}) \leq D_{\boldsymbol{\alpha}}(A',B';d_{\boldsymbol{\beta}})$ iff $D_{\boldsymbol{\alpha}'}(A,B;d_{\boldsymbol{\beta}'}) \leq D_{\boldsymbol{\alpha}'}(A',B';d_{\boldsymbol{\beta}'})$.*

*Proof sketch.* Let $\mathcal{H}$ be the linear functions constructed in Lemma 1 and fix any region $\mathcal{Z}$ in the sign-pattern partition induced by $\mathcal{H}$. For any $i \in [L']$, since the merge function $D_i$ is 2-point-based and the ordering over pairs of points according to $d_{\boldsymbol{\beta}}(\cdot,\cdot)$ in the region $\mathcal{Z}$ is fixed, for any clusters $A, B, A', B' \subset S$ we can find points $a_i, b_i, a_i', b_i'$ such that $D_i(A,B;d_{\boldsymbol{\beta}}) = d_{\boldsymbol{\beta}}(a_i,b_i)$ and $D_i(A',B';d_{\boldsymbol{\beta}}) = d_{\boldsymbol{\beta}}(a_i',b_i')$ for all $\boldsymbol{\beta} \in R$. Therefore, expanding the definition of $D_{\boldsymbol{\alpha}}$, we have that $D_{\boldsymbol{\alpha}}(A,B;d_{\boldsymbol{\beta}}) \leq D_{\boldsymbol{\alpha}}(A',B';d_{\boldsymbol{\beta}})$ iff $q_{A,B,A',B'}(\boldsymbol{\alpha},\boldsymbol{\beta}) \leq 0$, where $q_{A,B,A',B'}(\boldsymbol{\alpha},\boldsymbol{\beta}) = \sum_i \sum_j \alpha_i \beta_j (d_j(a_i,b_i) - d_j(a_i',b_i'))$. Observe that the coefficients of each quadratic function depend only on $4L'$ points in $S$, so there are only $O(|S|^{4L'})$ possible quadratics of this from collected across all regions $\mathcal{Z}$ in the sign-pattern partition induced by $\mathcal{H}$ and all subsets of 4 clusters. Together with $\mathcal{H}$, this set of quadratic functions partitions the joint parameter space into regions where the ordering over all pairs of clusters is fixed. $\qquad\square$

A consequence of Lemma 2 is that for any clustering instance $S$ with target clustering $\mathcal{Y}$, the function $(\boldsymbol{\alpha},\boldsymbol{\beta}) \mapsto \ell(\mathcal{A}_{\boldsymbol{\alpha},\boldsymbol{\beta}}(S),\mathcal{Y})$ is piecewise constant, where the constant partitioning is the sign-pattern partition induced by $O(|S|^{4L'})$ quadratic functions. The proof of Theorem 1 now follows from using this fact to bound the empirical Rademacher complexity of the learning problem.

**Extensions.** The above analysis can be extended to handle several more general settings. First, we can accommodate many specific merge functions that are not included in the 2-point-based family, at the cost of increasing the number of quadratic functions $|\mathcal{Q}|$ needed in Lemma 2. For example, if one of the merge functions is average linkage, $D_{avg}(A,B;d) = \frac{1}{|A|\cdot|B|}\sum_{a\in A, b \in B} d(a,b)$, then $|\mathcal{Q}|$ will be exponential in the dataset size $n$. Fortunately, our sample complexity analysis depends only on $\log(|\mathcal{Q}|)$, so this still leads to non-trivial sample complexity guarantees (though the computational algorithm selection problem becomes harder). We can also extend the analysis to more intricate methods for combining the metrics and merge functions. For example, our analysis applies to polynomial combinations of metrics and merges at the cost of increasing the complexity of the functions defining the piecewise constant partition.

## 3 EFFICIENT ALGORITHM SELECTION

In this section we provide efficient algorithms for learning low-loss clustering algorithms and metrics for application-specific distributions $\mathcal{D}$ defined over clustering instances. We begin by focusing on the special case where we have a single metric and our goal is to learn the best combination of two merge functions (i.e., $L = 1$ and $L' = 2$). This special case is already algorithmically interesting. Next, we show how to apply similar techniques to the case of learning the best combination of two metrics when using the complete linkage merge function (i.e., $L = 2$ and $L' = 1$). Finally, we discuss how to generalize our techniques to other cases.

**Learning the Merge Function.** We will use the following simplified notation for mixing two base merge functions $D_0(A,B;d)$ and $D_1(A,B;d)$: for each parameter $\alpha \in [0,1]$, let $D_{\alpha}(A,B;d) = (1-\alpha)D_0(A,B;d) + \alpha D_1(A,B;d)$ denote the convex combination with weight $(1-\alpha)$ on $D_0$ and weight $\alpha$ on $D_1$. We let $A_{\alpha}^{merge}(S;D_0,D_1)$ denote the cluster tree produced by the algorithm with parameter $\alpha$, and $\mathcal{A}_{merge}(D_0,D_1) = \{A_{\alpha}^{merge}(\cdot;D_0,D_1) \mid \alpha \in [0,1]\}$ denote the parameterized algorithm family.

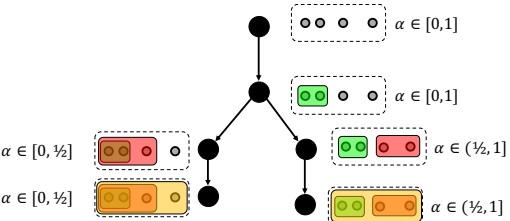

Figure 1: An example of the execution tree of $\mathcal{A}_{\text{merge}}(\mathrm{D_{min}}, \mathrm{D_{max}})$ for a clustering instance with 4 points. The nested rectangles show the clustering at each node.

Our goal is to design efficient procedures for finding the algorithm from $\mathcal{A}_{\text{merge}}(D_0, D_1)$ (and more general families) that has the lowest average loss on a sample of labeled clustering instances $(S_1, \mathcal{Y}_1), \ldots, (S_N, \mathcal{Y}_N)$ where $\mathcal{Y}_i = (C_1^{(i)}, \ldots, C_{k_i}^{(i)})$ is the target clustering for instance $S_i$. Recall that the loss function $\ell(T, \mathcal{Y})$ computes the Hamming distance between the target clustering $\mathcal{Y}$ and the closest pruning of the cluster tree $T$. Formally, our goal is to solve the following optimization problem: $\text{argmin}_{\alpha \in [0,1]} \frac{1}{N} \sum_{i=1}^{N} \ell(A_{\alpha}^{\text{merge}}(S_i; D_0, D_1), \mathcal{Y}_i)$.

The key challenge is that, for a fixed clustering instance $S$ we can partition the parameter space $[0, 1]$ into finitely many intervals such that for each interval $I$, the cluster tree output by the algorithm $A_{\alpha}^{\text{merge}}(S; D_0, D_1)$ is the same for every parameter in $\alpha \in I$. It follows that the loss function is a piecewise constant function of the algorithm parameter. Therefore, the optimization problem is non-convex and the loss derivative is zero wherever it is defined, rendering gradient descent and similar algorithms ineffective.

We solve the optimization problem by explicitly computing the piecewise constant loss function for each instance $S_i$. That is, for instance $i$ we find a collection of discontinuity locations $0 = c_0^{(i)} < \ldots < c_{M_i}^{(i)} = 1$ and values $v_1^{(i)}, \ldots, v_{M_i}^{(i)} \in \mathbb{R}$ so that for each $j \in [M_i]$, running the algorithm on instance $S_i$ with a parameter in $[c_{j-1}^{(i)}, c_j^{(i)})$ has loss equal to $v_j^{(i)}$. Given this representation of the loss function for each of the $N$ instances, finding the parameter with minimal average loss can be done in $O(M \log(M))$ time, where $M = \sum_i M_i$ is the total number of discontinuities from all $N$ loss functions. The bulk of the computational cost is incurred by computing the piecewise constant loss functions, which we focus on for the rest of the section.

We exploit a more powerful structural property of the algorithm family to compute the piecewise constant losses: for a clustering instance $S$ and any length $t$, the sequence of first $t$ merges performed by the algorithm is a piecewise constant function of the parameter (our sample complexity results only used that the final tree is piecewise constant). For length $t = 0$, the partition is a single region containing all parameters in $[0, 1]$, since every algorithm trivially starts with the empty sequence of merges. For each length $t > 0$, the piecewise constant partition for the first $t$ merges is a refinement of the partition for $t - 1$ merges. We can represent this sequence of partitions using a partition tree, where each node in the tree is labeled by an interval, the nodes at depth $t$ describe the partition of $[0, 1]$ after $t$ merges, and edges represent subset relationships. This tree represents all possible execution paths for the algorithm family when run on the instance $S$ as we vary the algorithm parameter. In particular, each path from the root node to a leaf corresponds to one possible sequence of merges. We therefore call this tree the *execution tree* of the algorithm family when run on $S$. Figure 1 shows an example execution tree for the family $\mathcal{A}_{\text{merge}}(\mathrm{D_{min}}, \mathrm{D_{max}})$. To find the piecewise constant loss function for a clustering instance $S$, it is sufficient to enumerate the leaves of the execution tree and compute the corresponding losses. The following result, proved in Appendix B, shows that the execution tree for $\mathcal{A}_{\text{merge}}(D_0, D_1)$ is well defined.

**Lemma 3.** *For any merge functions $\mathrm{D_0}$ and $\mathrm{D_1}$ and any clustering instance $S$, the execution tree for $\mathcal{A}_{\text{merge}}(\mathrm{D_0}, \mathrm{D_1})$ when run on $S$ is well defined. That is, there exists a partition tree s.t. for any node $v$ at depth $t$, the same sequence of first $t$ merges is performed by $A_{\alpha}^{\text{merge}}$ for all $\alpha$ in node $v$'s interval.*

The fundamental operation required to perform a depth-first traversal of the execution tree is finding a node's children. That is, given a node, its parameter interval $[\alpha_{\text{lo}}, \alpha_{\text{hi}})$, and the set of clusters at that node $C_1, \ldots, C_m$, find all possible merges that will be chosen by the algorithm for $\alpha \in [\alpha_{\text{lo}}, \alpha_{\text{hi}})$. We know that for each pair of merges $(C_i, C_j)$ and $(C_i', C_j')$, there is a single critical parameter value where the algorithm switches from preferring to merge $(C_i, C_j)$ to $(C_i', C_j')$. A direct algorithm that runs in $O(m^4)$ time for finding the children of a node in the execution tree is to compute all $O(m^4)$ critical parameter values and test which pair of clusters will be merged on each interval between

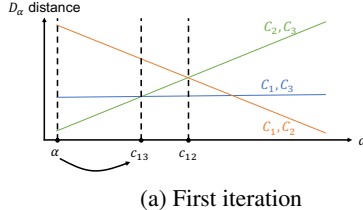
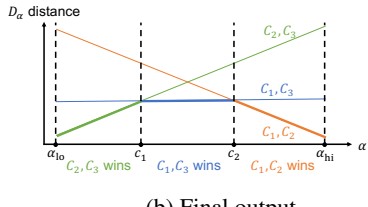

(a) First iteration                                        (b) Final output

Figure 2: Depiction of Algorithm 2 when given three clusters, $C_1$, $C_2$, and $C_3$. Each line shows the $D_\alpha$-distance between one pair of clusters as a function of the parameter $\alpha$. On the first iteration, Algorithm 2 determines that clusters $C_2$ and $C_3$ are the closest for the parameter $\alpha = \alpha_{\text{lo}}$, calculates the critical parameter values $c_{12}$ and $c_{13}$, and advances $\alpha$ to $c_{13}$. Repeating the process partitions of $[\alpha_{\text{lo}}, \alpha_{\text{hi}})$ into merge-constant regions.

consecutive critical parameters. We provide a more efficient algorithm that runs in time $O(m^2 M)$, where $M \leq m^2$ is the number of children of the node.

Fix any node in the execution tree. Given the node's parameter interval $I = [\alpha_{\text{lo}}, \alpha_{\text{hi}})$ and the set of clusters $C_1, \ldots, C_m$ resulting from that node's merge sequence, we use a sweep-line algorithm to determine all possible next merges and the corresponding parameter intervals. First, we calculate the merge for $\alpha = \alpha_{\text{lo}}$ by enumeration in $O(m^2)$ time. Suppose clusters $C_i$ and $C_j$ are the optimal merge for $\alpha$. We then determine the largest value $\alpha'$ for which $C_i$ and $C_j$ are still merged by solving the linear equation $D_\alpha(C_i, C_j) = D_\alpha(C_k, C_l)$ for all other pairs of clusters $C_k$ and $C_l$, keeping track of the minimal solution larger than $\alpha$. Since there are only $O(m^2)$ alternative pairs of clusters, this takes $O(m^2)$ time. Denote the minimal solution larger than $\alpha$ by $c \in I$. We are guaranteed that $A_{\alpha'}^{\text{merge}}$ will merge clusters $C_i$ and $C_j$ for all $\alpha' \in [\alpha, c)$. We repeat this procedure starting from $\alpha = c$ to determine the next merge and corresponding interval, and so on, sweeping through the $\alpha$ parameter space until $\alpha \geq \alpha_{\text{hi}}$. Algorithm 2 in Appendix B provides pseudocode for this approach and Figure 2 shows an example. Our next result bounds the running time of this procedure.

**Lemma 4.** *Let $C_1, \ldots, C_m$ be a collection of clusters, $\text{D}_0$ and $\text{D}_1$ be any pair of merge functions, and $[\alpha_{lo}, \alpha_{hi})$ be a subset of the parameter space. If there are $M$ distinct cluster pairs $C_i, C_j$ that minimize $\text{D}_\alpha(C_i, C_j)$ for values of $\alpha \in [\alpha_{lo}, \alpha_{hi})$, then the running time of Algorithm 2 is $O(Mm^2 K)$, where $K$ is the cost of evaluating the merge functions $\text{D}_0$ and $\text{D}_1$.*

With this, our algorithm for computing the piecewise constant loss function for an instance $S$ performs a depth-first traversal of the leaves of the execution tree for $\mathcal{A}_{\text{merge}}(\text{D}_0, \text{D}_1)$, using Algorithm 2 to determine the children of each node. When we reach a leaf in the depth-first traversal, we have both the corresponding parameter interval $I \subset [0, 1]$, as well as the cluster tree $T$ such that $A_\alpha^{\text{merge}}(S) = T$ for all $\alpha \in I$. We then evaluate the loss $\ell(T, \mathcal{Y})$ to get one piece of the piecewise constant loss function. Detailed pseudocode for this approach is given in Algorithm 3 in Appendix B.

**Theorem 2.** *Let $S = \{x_1, \ldots, x_n\}$ be a clustering instance and $\text{D}_0$ and $\text{D}_1$ be any two merge functions. Suppose that the execution tree of $\mathcal{A}_{merge}(\text{D}_0, \text{D}_1)$ on $S$ has $E$ edges. Then the total running time of Algorithm 3 is $O(En^2 K)$, where $K$ is the cost of evaluating $\text{D}_0$ and $\text{D}_1$ once.*

We can express the running time of Algorithm 3 in terms of the number of discontinuities of the function $\alpha \mapsto A_\alpha^{\text{merge}}(S)$. There is one leaf of the execution tree for each constant interval of this function, and the path from the root of the execution tree to that leaf is of length $n - 1$. Therefore, the cost associated with that path is at most $O(Kn^3)$ and enumerating the execution tree to obtain the piecewise constant loss function for a given instance $S$ spends $O(Kn^3)$ time for each constant interval of $\alpha \mapsto A_\alpha^{\text{merge}}(S)$. In contrast, the combinatorial approach of Balcan et al. (2017) requires that we run $\alpha$-linkage once for every interval in their partition of $[0, 1]$, which always contains $O(n^8)$ intervals (i.e., it is a refinement of the piecewise constant partition). Since each run of $\alpha$-Linkage costs $O(Kn^2 \log n)$ time, this leads to a running time of $O(Kn^{10} \log n)$. The key advantage of our approach stems from the fact that the number of discontinuities of the function $\alpha \mapsto A_\alpha^{\text{merge}}(S)$ is often several orders of magnitude smaller than $O(n^8)$.

**Learning the Metric.** Next we present efficient algorithms for computing the piecewise constant loss function for a single clustering instance when interpolating between two base metrics and using complete linkage. For a pair of fixed base metrics $d_0$ and $d_1$ and any parameter value $\beta \in [0, 1]$, define $d_\beta(a, b) = (1 - \beta)d_0(a, b) + \beta d_1(a, b)$. Let $A_\beta^{\text{metric}}(S; d_0, d_1)$ denote the output of running complete linkage with the metric $d_\beta$, and $\mathcal{A}_{\text{metric}}(d_0, d_1)$ denote the family of all such algorithms. We prove that for this algorithm family, the execution tree is well defined and provide an

efficient algorithm for finding the children of each node in the execution tree, allowing us to use a depth-first traversal to find the piecewise constant loss function for any clustering instance $S$.

**Lemma 5.** *For any metrics $\mathrm{d}_0$ and $\mathrm{d}_1$ and any clustering instance $S$, the execution tree for the family $\mathcal{A}_{metric}(\mathrm{d}_0, \mathrm{d}_1)$ when run on $S$ is well defined. That is, there exists a partition tree s.t. for any node $v$ at depth $t$, the same sequence of first $t$ merges is performed by $A_\beta^{metric}$ for all $\beta$ in node $v$'s interval.*

Next, we provide an efficient procedure for determining the children of a node $v$ in the execution tree of $\mathcal{A}_{\mathrm{metric}}(\mathrm{d}_0, \mathrm{d}_1)$. Given the node's parameter interval $I = [\beta_{\mathrm{lo}}, \beta_{\mathrm{hi}})$ and the set of clusters $C_1, \ldots, C_m$ resulting from that node's sequence of merges, we again use a sweep-line procedure to find the possible next merges and the corresponding parameter intervals. First, we determine the pair of clusters that will be merged by $A_\beta^{\mathrm{metric}}$ for $\beta = \beta_{\mathrm{lo}}$ by enumerating all pairs of clusters. Suppose the winning pair is $C_i$ and $C_j$ and let $x \in C_i$ and $x' \in C_j$ be the farthest pair of points between the two clusters. Next, we find the largest value of $\beta'$ for which we will still merge the clusters $C_i$ and $C_j$. To do this, we enumerate all other pairs of clusters $C_k$ and $C_l$ and all pairs of points $y \in C_k$ and $y' \in C_l$, and solve the linear equation $\mathrm{d}_{\beta'}(x, x') = \mathrm{d}_\beta(y, y')$, keeping track of the minimal solution larger than $\beta$. Denote the minimal solution larger than $\beta$ by $c$. We are guaranteed that for all $\beta' \in [\beta, c)$, the pair of clusters merged will be $C_i$ and $C_j$. Then we repeat the process with $\beta = c$ to find the next merge and corresponding interval, and so on, until $\beta \geq \beta_{\mathrm{hi}}$. Pseudocode for this procedure is given in Algorithm 4 in Appendix B. The following Lemma bounds the running time:

**Lemma 6.** *Let $C_1, \ldots, C_m$ be a collection of clusters, $\mathrm{d}_0$ and $\mathrm{d}_1$ be any pair of metrics, and $[\beta_{lo}, \beta_{hi})$ be a subset of the parameter space. If there are $M$ distinct cluster pairs $C_i, C_j$ that complete linkage would merge when using the metric $\mathrm{d}_\beta$ for $\beta \in [\beta_{lo}, \beta_{hi})$, the running time of Algorithm 4 is $O(Mn^2)$.*

Our algorithm for computing the piecewise constant loss function for an instance $S$ is almost identical for the case of the merge function: it performs a depth-first traversal of the leaves of the execution tree for $\mathcal{A}_{\mathrm{metric}}(\mathrm{d}_0, \mathrm{d}_1)$, using Algorithm 4 to determine the children of each node. Detailed pseudocode for this approach is given in Algorithm 5 in Appendix B. The following Theorem characterizes the overall running time of the algorithm.

**Theorem 3.** *Let $S = \{x_1, \ldots, x_n\}$ be a clustering instance and $\mathrm{d}_0$ and $\mathrm{d}_1$ be any two merge functions. Suppose that the execution tree of $\mathcal{A}_{metric}(\mathrm{d}_0, \mathrm{d}_1)$ on $S$ has $E$ edges. Then the total running time of Algorithm 5 is $O(En^2)$.*

**Extension to general algorithm families.** In this section we introduced efficient algorithm selection procedures for two special cases where the algorithm family has a single parameter. Working with a one-dimensional parameter space allowed us to compute partitions of the parameter space using efficient sweep-line procedures. The key additional challenge when working with higher-dimensional parameter spaces is that the partition of the parameter space given by each level of the execution tree becomes more complicated, defined by the sign-pattern partition induced by a collection of quadratic functions. Efficient execution-tree based algorithm selection procedures for our more general algorithm family will require an analog of the sweep-line algorithms described above that efficiently compute the children of a node in the execution tree, together with a minimal number of quadratic functions describing the partition.

It may also be promising to explore combinatorial-style algorithm selection procedures. As a consequence of Lemma 2, for any collection of $N$ clustering instances with at most $n$ points, we can find $O(Nn^{4L'})$ quadratic functions whose sign-pattern partition divides the parameter space into polynomially many regions where the algorithm output is constant on every instance. It follows that we can find an empirically optimal parameter in polynomial time if we are able to identify a parameter from each region of the sign-pattern partition in polynomial time. For example, when the parameter space is $\mathbb{R}^2$, for each vertex $v$ of each region in the sign-pattern partition, there is a pair of quadratic functions such that $q_i(v) = q_j(v) = 0$. By solving this equation for all pairs of quadratic functions, we are able to find a parameter from each region. An interesting future direction is to develop efficient analogs for higher-dimensional parameter spaces.

## 4    EXPERIMENTS

In this section we evaluate the performance of our learning procedures when finding algorithms for application-specific clustering distributions. Our experiments demonstrate that the best algorithm for different applications varies greatly, and that in many cases we can have large gains in cluster quality using a mixture of base merge functions or metrics.

**Experimental setup.** In each experiment we define a distribution $\mathcal{D}$ over clustering tasks. For each clustering instance, the loss of the cluster tree output by a clustering algorithm is measured in terms of the loss $\ell(S, \mathcal{Y})$, which computes the Hamming distance between the target clustering and the closest pruning of the cluster tree. We draw $N$ sample

clustering tasks from the given distribution and use the algorithms developed in Section 3 to exactly compute the average empirical loss for every algorithm in one algorithm family. The theoretical results from Section 2 ensure that these plots generalize to new samples from the same distribution, so our focus is on demonstrating empirical improvements in clustering loss obtained by learning the merge function or metric.

**Clustering distributions.** Most of our clustering distributions are generated from classification datasets by sampling a subset of the dataset and using the class labels as the target clustering. We briefly describe our instance distributions together with the metrics used for each below. Complete details for the distributions can be found in Appendix C.

*MNIST Subsets.* The MNIST dataset (LeCun et al., 1998) contains images of hand-written digits from 0 to 9. We generate a random clustering instance from this data by choosing $k = 5$ random digits and sampling 200 images from each digit, giving a total of $n = 1000$ images. We measure distance between any pairs of images using the Euclidean distance between their pixel intensities.

*CIFAR-10 Subsets.* We similarly generate clustering instances from the CIFAR-10 dataset (Krizhevsky, 2009). To generate an instance, we select $k = 5$ classes at random and then sample 50 images from each class, leading to a total of $n = 250$ images. We measure distances between examples using the cosine distance between feature embedding extracted from a pre-trained Google inception network (Szegedy et al., 2015).

*Omniglot Subsets.* The Omniglot dataset (Lake et al., 2015) contains written characters from 50 alphabets, with a total of 1623 different characters. To generate a clustering instance from the omniglot data, we choose one of the alphabets at random, we sample $k$ from $\{5, \ldots, 10\}$ uniformly at random, choose $k$ random characters from the alphabet, and include all 20 examples of those characters in the clustering instance. We use two metrics for the omniglot data: first, cosine distances between neural network feature embeddings of the character images from a simplified version of AlexNet (Krizhevsky et al., 2012). Second, each character is also described by a "stroke", which is a sequence of coordinates $(x_t, y_t)_{t=1}^T$ describing the trajectory of the pen when writing the character. We hand-design a metric based on the stroke data: the distance between a pair of characters is the average distance from a point on either stroke to its nearest neighbor on the other stroke. A formal definition is given in the appendix.

*Places2 Subsets.* The Places2 dataset consists of images of 365 different place categories, including "volcano", "gift shop", and "farm" (Zhou et al., 2017). To generate a clustering instance from the places data, we choose $k$ randomly from $\{5, \ldots, 10\}$, choose $k$ random place categories, and then select 20 random examples from each chosen category. We use two metrics for this data distribution. First, we use cosine distances between feature embeddings generated by a VGG16 network (Simonyan and Zisserman, 2015) pre-trained on ImageNet (Deng et al., 2009). Second, we compute color histograms in HSV space for each image and use the cosine distance between the histograms.

*Places2 Diverse Subsets.* We also construct an instance distribution from a subset of the Places2 classes which have diverse color histograms. We expect the color histogram metric to perform better on this distribution. To generate a clustering instance, we pick $k = 4$ classes from aquarium, discotheque, highway, iceberg, kitchen, lawn, stage-indoor, underwater ocean deep, volcano, and water tower. We include 50 randomly sampled images from each chosen class, leading to a total of $n = 200$ points per instance.

*Synthetic Rings and Disks.* We consider a two dimensional synthetic distribution where each clustering instance has 4 clusters, where two are ring-shaped and two are disk-shaped. To generate each instance we sample 100 points uniformly at random from each ring or disk. The two rings have radiuses $0.4$ and $0.8$, respectively, and are both centered at the origin. The two disks have radius $0.4$ and are centered at $(1.5, 0.4)$ and $(1.5, -0.4)$, respectively. For this data, we measure distances between points in terms of the Euclidean distance between them.

**Results.** *Learning the Merge Function.* Figure 3 shows the average loss when interpolating between single and complete linkage as well as between average and complete linkage for each of the clustering instance distributions described above. For each value of the parameter $\alpha \in [0, 1]$, we report the average loss over $N = 1000$ i.i.d. instances drawn from the corresponding distribution. We see that the optimal parameters vary across different clustering instances. For example, when interpolating between single and complete linkage, the optimal parameters are $\alpha = 0.874$ for MNIST, $\alpha = 0.98$ for CIFAR-10, $\alpha = 0.179$ for Rings and Disks, and $\alpha = 0.931$ for Omniglot. Moreover, using the parameter that is optimal for one distribution on another would lead to significantly worse clustering performance. Next, we also see that for different distributions, it is possible to achieve non-trivial improvements over single, complete, and average linkage by interpolating between them. For example, on the Rings and Disks distribution we see an improvement of almost 0.2 error, meaning that an additional 20% of the data is correctly clustered.

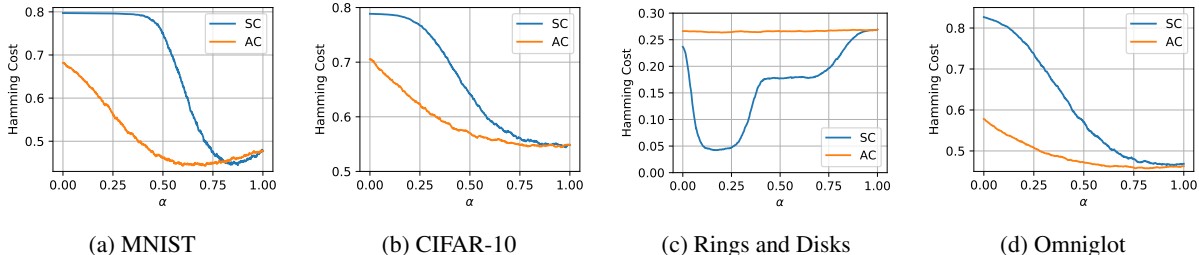

(a) MNIST       (b) CIFAR-10       (c) Rings and Disks       (d) Omniglot

Figure 3: Empirical loss for interpolating between single and complete linkage ('SC' in the legend) as well as between average and complete linkage ('AC' in the legend) over 1000 sampled clustering instances.

*Learning the Metric.* Next we consider learning the best metric for the Omniglot, Places2, and Places2 Diverse instance distributions. Each of these datasets is equipped with one hand-designed metric and one metric based on neural-network embeddings. The parameter $\beta = 0$ corresponds to the hand-designed metric, while $\beta = 1$ corresponds to the embedding. Figure 4 shows the empirical loss for each parameter $\beta$ averaged over $N = 4000$ samples for each distribution. On all three distributions the neural network embedding performs better than the hand-designed metric, but we can achieve non-trivial performance improvements by mixing the two metrics. On Omniglot, the optimal parameter is at $\beta = 0.514$ which improves the Hamming error by $0.091$, meaning that we correctly cluster nearly $10\%$ more of the data. For the Places2 distribution we see an improvement of approximately $1\%$ with the optimal parameter being $\beta = 0.88$, while for the Places2 Diverse distribution the improvement is approximately $3.4\%$ with the optimal $\beta$ being $0.87$.

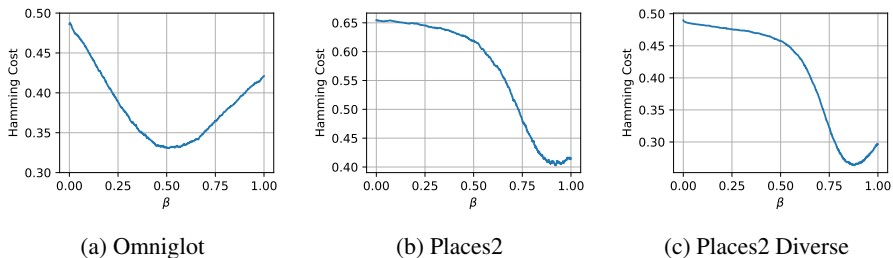

(a) Omniglot       (b) Places2       (c) Places2 Diverse

Figure 4: Empirical loss interpolating between two distance metrics on Omniglot, Places2, and Places2 distributions. In each plot, $\beta = 0$ corresponds to the hand-crafted metric and $\beta = 1$ corresponds to the neural network embedding.

*Number of Discontinuities.* The efficiency of our algorithm selection procedures stems from the fact that their running time scales with the true number of discontinuities in each loss function, rather than a worst-case upper bound. Of all the experiments we ran, interpolating between single interpolating between single and complete linkage for MNIST had the most discontinuities per loss function with an average of $362.6$ discontinuities per function. Given that these instances have $n = 1000$ points, this leads to a speedup of roughly $n^8/362.6 \approx 2.76 \times 10^{21}$ over the combinatorial algorithm that solves for all $O(n^8)$ critical points and runs the clustering algorithm once for each. Table 1 in Appendix C shows the average number of discontinuities per loss function for all of the above experiments.

## 5   CONCLUSION

In this work we study both the sample and algorithmic complexity of learning linkage-based clustering algorithms with low loss for specific application domains. We give strong bounds on the number of sample instances required from an application domain in order to find an approximately optimal algorithm from a rich family of algorithms that allows us to vary both the metric and merge function used by the algorithm. We complement our sample complexity results with efficient algorithms for finding empirically optimal algorithms for a sample of instances. Finally, we carry out experiments on both real-world and synthetic clustering domains demonstrating that our procedures can often find algorithms that significantly outperform standard linkage-based clustering algorithms. An important future direction for learning linkage based clustering algorithms is the design of efficient algorithm selection procedures for higher dimensional parameter spaces.

ACKNOWLEDGEMENTS

This work was supported in part by NSF grants CCF-1535967, IIS-1618714, IIS-1901403, CCF-1910321, SES-1919453, an Amazon Research Award, a Bloomberg Research Grant, a Microsoft Research Faculty Fellowship, and by the generosity of Eric and Wendy Schmidt by recommendation of the Schmidt Futures program.

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

# A  APPENDIX FOR LEARNING CLUSTERING ALGORITHMS

We begin by providing complete proofs for the piecewise structural Lemmas from the main body.

**Lemma 1.** *Fix any metrics $d_1, \ldots, d_L$ and a clustering instance $S \subset \mathcal{X}$. There exists a set $\mathcal{H}$ of $O(|S|^4)$ linear functions mapping $\mathbb{R}^L$ to $\mathbb{R}$ with the following property: if two metric parameters $\boldsymbol{\beta}, \boldsymbol{\beta}' \in \Delta_L$ belong to the same region in the sign-pattern partition induced by $\mathcal{H}$, then the ordering over pairs of points in $S$ given by $d_{\boldsymbol{\beta}}$ and $d_{\boldsymbol{\beta}'}$ are the same. That is, for all points $a, b, a', b' \in S$ we have $d_{\boldsymbol{\beta}}(a,b) \leq d_{\boldsymbol{\beta}}(a',b')$ iff $d_{\boldsymbol{\beta}'}(a,b) \leq d_{\boldsymbol{\beta}'}(a',b')$.*

*Proof.* Let $S$ be any clustering instance and fix points $a, b, a', b' \in S$. For any parameter $\boldsymbol{\beta} \in \Delta_L$, by definition of $d_{\boldsymbol{\beta}}$, we have that

$$d_{\boldsymbol{\beta}}(a,b) \leq d_{\boldsymbol{\beta}}(a',b') \iff \sum_{i=1}^{L} \beta_i d_i(a,b) \leq \sum_{i=1}^{L} \beta_i d_i(a',b') \iff \sum_{i=1}^{L} \beta_i(d_i(a,b) - d_i(a',b')) \leq 0.$$

Define the linear function $h_{a,b,a',b'}(\boldsymbol{\beta}) = \sum_{i=1}^{L} \beta_i(d_i(a,b) - d_i(a',b'))$. Then we have that $d_{\boldsymbol{\beta}}(a,b) \leq d_{\boldsymbol{\beta}}(a',b')$ if $h_{a,b,a',b'}(\boldsymbol{\beta}) \leq 0$ and $d_{\boldsymbol{\beta}}(a,b) > d_{\boldsymbol{\beta}}(a',b')$ if $h_{a,b,a',b'}(\boldsymbol{\beta}) > 0$.

Let $\mathcal{H} = \{h_{a,b,a',b'} \mid a, b, a', b' \in S\}$ be the collection of all such linear functions collected over all possible subsets of 4 points in $S$. Now suppose that $\boldsymbol{\beta}$ and $\boldsymbol{\beta}'$ belong to the same region in the sign-pattern partition induced by $\mathcal{H}$. For any points $a, b, a', b' \in S$, we are guaranteed that $\text{sign}(h_{a,b,a',b'}(\boldsymbol{\beta})) = \text{sign}(h_{a,b,a',b'}(\boldsymbol{\beta}'))$, which by the above arguments imply that $d_{\boldsymbol{\beta}}(a,b) \leq d_{\boldsymbol{\beta}}(a',b')$ iff $d_{\boldsymbol{\beta}'}(a,b) \leq d_{\boldsymbol{\beta}'}(a',b')$, as required. $\square$

**Lemma 2.** *Fix any metrics $d_1, \ldots, d_L$, any 2-point-based merge functions $D_1, \ldots, D_{L'}$, and clustering instance $S \subset \mathcal{X}$. There exists a set $\mathcal{Q}$ of $O(|S|^{4L'})$ quadratic functions defined on $\mathbb{R}^{L'+L}$ so that if parameters $(\boldsymbol{\alpha}, \boldsymbol{\beta})$ and $(\boldsymbol{\alpha}', \boldsymbol{\beta}')$ belong to the same region of the sign-pattern partition induced by $\mathcal{Q}$, then the ordering over pairs of clusters in $S$ given by $D_{\boldsymbol{\alpha}}(\cdot, \cdot; d_{\boldsymbol{\beta}})$ and $D_{\boldsymbol{\alpha}'}(\cdot, \cdot; d_{\boldsymbol{\beta}'})$ is the same. That is, for all clusters $A, B, A', B' \subset S$, we have that $D_{\boldsymbol{\alpha}}(A, B; d_{\boldsymbol{\beta}}) \leq D_{\boldsymbol{\alpha}}(A', B'; d_{\boldsymbol{\beta}})$ iff $D_{\boldsymbol{\alpha}'}(A, B; d_{\boldsymbol{\beta}'}) \leq D_{\boldsymbol{\alpha}'}(A', B'; d_{\boldsymbol{\beta}'})$.*

*Proof.* From Lemma 1, we know we can find a set $\mathcal{H}$ of $O(|S|^4)$ linear functions defined on $\mathbb{R}^L$ that induce a sign-pattern partition of the $\boldsymbol{\beta}$ parameter space $\Delta_L \subset \mathbb{R}^L$ into regions where the ordering over pairs of points according to the $d_{\boldsymbol{\beta}}$ distance is constant.

Now let $\mathcal{Z} \subset \Delta_L$ be any region of the sign-pattern partition of $\Delta_L$ induced by $\mathcal{H}$. From Lemma 1, we know that for all parameters $\boldsymbol{\beta} \in \mathcal{Z}$, the ordering over pairs of points in $S$ according to $d_{\boldsymbol{\beta}}$ is fixed. For any 2-point-based merge function, the pair of points used to measure the distance between a pair of clusters depends only on the ordering of pairs of points according to distance. Therefore, since $D_1, \ldots, D_{L'}$ are all 2-point-based, we know that for any pair of clusters $(A, B)$ and each merge function index $i \in [L']$, there exists a pair of points $(a_i, b_i) \in A \times B$ such that $D_i(A, B; d_{\boldsymbol{\beta}}) = d_{\boldsymbol{\beta}}(a_i, b_i)$ for all $\boldsymbol{\beta} \in \mathcal{Z}$. In other words, all of the merge functions measure distances between $A$ and $B$ using a fixed pair of points for all values of the metric parameter $\boldsymbol{\beta}$ in the region $\mathcal{Z}$. Similarly, let $A', B' \subset S$ be any other pair of clusters and $(a_i', b_i') \in A' \times B'$ be the pairs of points defining $D_i(A', B'; d_{\boldsymbol{\beta}})$ for each $i \in [L']$. Then for

all $\boldsymbol{\beta} \in \mathcal{Z}$, we have that

$$D_{\boldsymbol{\alpha}}(A, B; d_{\boldsymbol{\beta}}) \leq D_{\boldsymbol{\alpha}}(A', B'; d_{\boldsymbol{\beta}}) \iff \sum_{i=1}^{L'} \alpha_i D_i(A, B; d_{\boldsymbol{\beta}}) \leq \sum_{i=1}^{L'} \alpha_i D_i(A, B; d_{\boldsymbol{\beta}})$$

$$\iff \sum_{i=1}^{L'} \alpha_i \sum_{j=1}^{L} \beta_j d_j(a_i, b_i) \leq \sum_{i=1}^{L'} \alpha_i \sum_{j=1}^{L} \beta_j d_j(a_i', b_i')$$

$$\iff \sum_{i=1}^{L'} \sum_{j=1}^{L} \alpha_i \beta_j \big( d_j(a_i, b_i) - d_j(a_i', b_i') \big) \leq 0.$$

Now define the quadratic function

$$q_{A,B,A',B'}(\boldsymbol{\alpha}, \boldsymbol{\beta}) = \sum_{i=1}^{L'} \sum_{j=1}^{L} \alpha_i \beta_j \big( d_j(a_i, b_i) - d_j(a_i', b_i') \big). \tag{1}$$

For all $\boldsymbol{\beta} \in \mathcal{Z}$, we are guaranteed that $D_{\boldsymbol{\alpha}}(A, B; d_{\boldsymbol{\beta}}) \leq D_{\boldsymbol{\alpha}}(A', B'; d_{\boldsymbol{\beta}})$ if and only if $q_{A,B,A',B'}(\boldsymbol{\alpha}, \boldsymbol{\beta}) \leq 0$. Notice that the coefficients of $q_{A,B,A',B'}$ only depend on $4L'$ points in $S$, which implies that if we collect these quadratic functions over all quadruples of clusters $A, B, A', B' \subset S$, we will only obtain $O(|S|^{4L'})$ different quadratic functions. These $O(|S|^{4L'})$ functions induce a sign-pattern partition of $\Delta_{L'} \times \mathcal{Z}$ for which the desired conclusion holds. Next, observe that the coefficients in the quadratic functions defined above do not depend on the region $\mathcal{Z}$ we started with. It follows that the same set of $O(|S|^{4L'})$ quadratic functions partition any other region $\mathcal{Z}'$ in the sign-pattern partition induced by $\mathcal{H}$ so that the claim holds on $\Delta_{L'} \times \mathcal{Z}'$.

Now let $\mathcal{Q}$ contain the linear functions in $\mathcal{H}$ (viewed as quadratic functions over $\mathbb{R}^{L'+L}$ by placing a zero coefficient on all quadratic terms and terms depending on $\boldsymbol{\alpha}$), together with the $O(|S|^{4L'})$ quadratic functions defined above. Then we have that $|\mathcal{Q}| = O(|S|^4 + |S|^{4L'}) = O(|S|^{4L'})$. Now suppose that $(\boldsymbol{\alpha}, \boldsymbol{\beta})$ and $(\boldsymbol{\alpha}', \boldsymbol{\beta}')$ belong to the same region of the sign-pattern partition of $\Delta_{L'} \times \Delta_L \subset \mathbb{R}^{L'+L}$ induced by the quadratic functions $\mathcal{Q}$. Since $\mathcal{Q}$ contains $\mathcal{H}$, this implies that $\boldsymbol{\beta}$ and $\boldsymbol{\beta}'$ belong to the same region $\mathcal{Z}$ in the sign-pattern partition induced by $\mathcal{H}$. Moreover, since $\mathcal{Q}$ contains all the quadratic functions defined in (1), it follows that $D_{\boldsymbol{\alpha}}(A, B; d_{\boldsymbol{\beta}}) \leq D_{\boldsymbol{\alpha}}(A', B'; d_{\boldsymbol{\beta}})$ if and only if $D_{\boldsymbol{\alpha}'}(A, B; d_{\boldsymbol{\beta}'}) \leq D_{\boldsymbol{\alpha}'}(A', B'; d_{\boldsymbol{\beta}'})$, as required. $\qquad\square$

Our proof of Theorem 1 depends crucially on the number of regions in the sign-pattern partition induced by a collection of functions. Based on the work of Buck (1943), we know that when the functions $f_1, \ldots, f_M$ are *linear*, then the number of regions is at most $(3M)^p$—a substantial improvement over the naive bound of $2^M$. We can also leverage this result to bound the number of regions by $(3M)^{p^2+p}$ when the functions are *quadratic* by viewing each quadratic function as a linear function on a $(p^2 + p)$-dimensional space. The following result summarizes these facts.

**Lemma 7.** *Let $T$ be the number of regions in the sign-pattern partition of $\mathbb{R}^p$ induced by $f_1, \ldots, f_M : \mathbb{R}^p \to \mathbb{R}$. The following statements hold.*

1. *If the functions are linear, i.e., $f_i(\boldsymbol{\zeta}) = \boldsymbol{w}_i^\top \boldsymbol{\zeta} + r_i$ for $\boldsymbol{w}_i \in \mathbb{R}^p$ and $r_i \in \mathbb{R}$, then $T \leq (3M)^p$.*

2. *If the functions are quadratic, i.e., $f_i(\boldsymbol{\zeta}) = \boldsymbol{\zeta}^\top Q_i \boldsymbol{\zeta} + \boldsymbol{w}_i^\top \boldsymbol{\zeta} + r_i$ for $Q_i \in \mathbb{R}^{p \times p}$, $\boldsymbol{w}_i \in \mathbb{R}^p$ and $r_i \in \mathbb{R}$, then $T \leq (3M)^{p^2+p}$.*

*Proof.* The proof of the first statement follows from the work of Buck (1943) which shows that if $\mathcal{H}$ is a collection of $M$ hyperplanes in $\mathbb{R}^p$, then the number of connected components of $\mathbb{R}^p \setminus \mathcal{H}$ is at most $(3M)^p$. To connect this to the sign-pattern partitioning induced by the collection of linear functions $f_1, \ldots, f_M$, let $\mathcal{H}$ be the set of $M$ hyperplanes defined by $\{\boldsymbol{\zeta} \in \mathbb{R}^p \mid f_i(\boldsymbol{\zeta}) = 0\}$ for $i \in [M]$. The connected components of $\mathbb{R}^p \setminus \mathcal{H}$ correspond exactly to the sign-pattern partition of the functions $f_1, \ldots, f_M$, and by the result of Buck (1943), it follows that the number of regions in the partition is at most $(3M)^p$. Equivalently, we know that $|\{F(\boldsymbol{\zeta}) \mid \zeta \in \mathbb{R}^p\}| \leq (3M)^p$, where $F(\boldsymbol{\zeta}) = \big(\text{sign}(f_1(\boldsymbol{\zeta})), \ldots, \text{sign}(f_M(\boldsymbol{\zeta}))\big)$ is the sign-pattern function. That is, $F$ takes at most $(3M)^p$ distinct values.

Next we use the first statement to prove the second statement. Let $\varphi : \mathbb{R}^p \to \mathbb{R}^{p^2+p}$ be the function that maps a vector $\boldsymbol{\zeta}$ to the vector containing all products of pairs of elements from $\boldsymbol{\zeta}$ concatenated to the beginning of $\boldsymbol{\zeta}$:

$$\varphi(\boldsymbol{\zeta}) = (\zeta_1 \zeta_1, \ldots, \zeta_1 \zeta_d, \ \ldots, \ \zeta_d \zeta_1, \ldots, \zeta_d \zeta_d, \ \zeta_1, \ldots, \zeta_d).$$

Let $q(\boldsymbol{\zeta}) = \boldsymbol{\zeta}^\top Q \boldsymbol{\zeta} + \boldsymbol{w}^\top \boldsymbol{\zeta} + r$ be a quadratic function where $Q \in \mathbb{R}^{p \times p}$, $\boldsymbol{w} \in \mathbb{R}^p$, and $r \in \mathbb{R}$. Now let

$$\boldsymbol{v} = (q_{11}, \ldots, q_{1d}, \ \ldots, \ q_{d1}, \ldots, q_{dd}, \ w_1, \ldots, w_d).$$

Then we have that $q(\boldsymbol{\zeta}) = \boldsymbol{v}^\top \varphi(\boldsymbol{\zeta})$. In other words, $q$ is a linear function of $\varphi(\boldsymbol{\zeta})$. This guarantees that we can find $M$ linear functions $h_1, \ldots, h_M : \mathbb{R}^{p^2+p} \to \mathbb{R}$ such that $f_i(\boldsymbol{\zeta}) = h_i(\varphi(\boldsymbol{\zeta}))$ for all $i \in [M]$. Let $H : \mathbb{R}^{p^2+p} \to \{\pm 1\}^M$ be the sign-pattern function for $h_1, \ldots, h_M$. Then we have

$$\left| \{ F(\boldsymbol{\zeta}) \mid \boldsymbol{\zeta} \in \mathbb{R}^p \} \right| = \left| \{ H(\varphi(\boldsymbol{\zeta})) \mid \boldsymbol{\zeta} \in \mathbb{R}^p \} \right| \leq \left| \{ H(\boldsymbol{\phi}) \mid \boldsymbol{\phi} \in \mathbb{R}^{p^2+p} \} \right| \leq (3M)^{p^2+p},$$

where the last inequality follows from the first statement for linear functions. It follows that the number of regions in the sign-pattern partition induced by $M$ quadratic functions is at most $(3M)^{p^2+p}$. $\qquad \square$

With this, we are ready to prove Theorem 1.

**Theorem 1.** *Fix any metrics $d_1, \ldots, d_L$, 2-point-based merge functions $D_1, \ldots, D_{L'}$, and distribution $\mathcal{D}$ over clustering instances with at most $n$ points. For any parameters $\epsilon > 0$ and $\delta > 0$, let $(S_1, \mathcal{Y}_1), \ldots, (S_N, \mathcal{Y}_N)$ be an i.i.d. sample of $N = O\left( \frac{1}{\epsilon^2} \left( (L' + L)^2 L' \log\left( \frac{(L'+L)^2 L' n}{\epsilon^2} \right) + \log\left( \frac{1}{\delta} \right) \right) \right) = \tilde{O}\left( \frac{(L'+L)^2 L'}{\epsilon^2} \right)$ clustering instances with target clusterings drawn from $\mathcal{D}$. Then with probability at least $1 - \delta$ over the draw of the sample, we have*

$$\sup_{(\boldsymbol{\alpha}, \boldsymbol{\beta}) \in \Delta_{L'} \times \Delta_L} \left| \frac{1}{N} \sum_{i=1}^N \ell(\mathcal{A}_{\boldsymbol{\alpha}, \boldsymbol{\beta}}(S_i), \mathcal{Y}_i) - \mathop{\mathbb{E}}_{(S, \mathcal{Y}) \sim \mathcal{D}} \left[ \ell(\mathcal{A}_{\boldsymbol{\alpha}, \boldsymbol{\beta}}(S), \mathcal{Y}) \right] \right| \leq \epsilon.$$

*Proof of Theorem 1.* Our goal is to provide a uniform convergence guarantee that ensures the sample loss of any pair of parameters $(\boldsymbol{\alpha}, \boldsymbol{\beta})$ is close to its expected cost on the underlying distribution. Towards that end, define the family of loss functions $\mathcal{F} = \{ f_{\boldsymbol{\alpha}, \boldsymbol{\beta}} : (S, \mathcal{Y}) \mapsto \ell(\mathcal{A}_{\boldsymbol{\alpha}, \boldsymbol{\beta}}(S), \mathcal{Y}) \mid (\boldsymbol{\alpha}, \boldsymbol{\beta} \in \Delta_{L'} \times \Delta_L) \}$, where the function $f_{\boldsymbol{\alpha}, \boldsymbol{\beta}}$ fixes algorithm parameters and maps each clustering instance $S$ with target clustering $\mathcal{Y}$ to that algorithm's loss when run on $(S, \mathcal{Y})$. We will bound the empirical Rademacher complexity of $\mathcal{F}$ to prove uniform convergence guarantees. Let $\mathcal{S} = \{ (S_1, \mathcal{Y}_1), \ldots, (S_N, \mathcal{Y}_N) \}$ be a sample of clustering instances with target clusterings drawn from the distribution $\mathcal{D}$. The empirical Rademacher complexity of the class of functions $\mathcal{F}$ on the sample $\mathcal{S}$ is given by

$$\hat{R}(\mathcal{F}, \mathcal{S}) = \frac{1}{N} \mathop{\mathbb{E}}_{\boldsymbol{\sigma}} \left[ \sup_{f_{\boldsymbol{\alpha}, \boldsymbol{\beta}} \in \mathcal{F}} \sum_{i=1}^N \sigma_i f_{\boldsymbol{\alpha}, \boldsymbol{\beta}}(S_i, \mathcal{Y}_i) \right],$$

where $\boldsymbol{\sigma}$ is a vector of $N$ independent Rademacher random variables.

From Lemma 2, we know that we can find a collection of $O(n^{4L'})$ quadratic functions for each clustering instance $S_i$ so that the output of Algorithm 1 when run on $S_i$ is constant on each region of the induced sign-pattern partition of the parameter space. Let $\mathcal{Q}$ be the collection of $O(Nn^{4L'})$ quadratic functions collected across all $N$ clustering instances $S_1, \ldots, S_N$. Whenever two parameter settings $(\boldsymbol{\alpha}, \boldsymbol{\beta})$ and $(\boldsymbol{\alpha}', \boldsymbol{\beta}')$ belong to the same region in the sign-pattern partition induced by $\mathcal{Q}$, we know that for all instance indices $i$, we have $\ell_{\boldsymbol{\alpha}, \boldsymbol{\beta}}(S_i, \mathcal{Y}_i) = \ell_{\boldsymbol{\alpha}', \boldsymbol{\beta}'}(S_i, \mathcal{Y}_i)$. Applying the second statement of Lemma 7, we know that this partition has at most $T = (3|\mathcal{Q}|)^{(L'+L)^2 + (L'+L)}$ regions. This implies that we can find a collection of $T$ parameter vectors $(\boldsymbol{\alpha}_1, \boldsymbol{\beta}_1), \ldots, (\boldsymbol{\alpha}_T, \boldsymbol{\beta}_T)$ (by taking one from each region in the partition) so that the following holds: for every $(\boldsymbol{\alpha}, \boldsymbol{\beta}) \in \Delta_{L'} \times \Delta_L$, there exists a $j \in [T]$ such that for all clustering instances $(S_i, \mathcal{Y}_i)$ in the sample $\mathcal{S}$ we have $f_{\boldsymbol{\alpha}, \boldsymbol{\beta}}(S_i, \mathcal{Y}_i) = f_{\boldsymbol{\alpha}_j, \boldsymbol{\beta}_j}(S_i, \mathcal{Y}_i)$. With this, we can bound the empirical Rademacher complexity as follows:

$$\hat{R}(\mathcal{F}, \mathcal{S}) = \frac{1}{N} \mathop{\mathbb{E}}_{\boldsymbol{\sigma}} \left[ \sup_{f_{\boldsymbol{\alpha}, \boldsymbol{\beta}} \in \mathcal{F}} \sum_{i=1}^N \sigma_i f_{\boldsymbol{\alpha}, \boldsymbol{\beta}}(S_i, \mathcal{Y}_i) \right] = \frac{1}{N} \mathop{\mathbb{E}}_{\boldsymbol{\sigma}} \left[ \sup_{1 \leq j \leq T} \sum_{i=1}^N \sigma_i f_{\boldsymbol{\alpha}_j, \boldsymbol{\beta}_j}(S_i, \mathcal{Y}_i) \right].$$

Now, using the fact that the losses take values in $[0, 1]$ and the supremum is over only $T$ distinct elements, we can apply Massart's Lemma (Massart, 2000) to obtain $\hat{R}(\mathcal{F}, \mathcal{S}) = O(\sqrt{\log(T)/N})$. Taking logs and simplifying our bound on $T$,

we have that $\log(T) = O\big((L' + L)^2 L' \log(Nn)\big)$, giving the following bound on the empirical Rademacher complexity of $\mathcal{F}$:

$$\hat{R}(\mathcal{F}, \mathcal{S}) = O\left(\sqrt{\frac{(L' + L)^2 L' \log(Nn)}{N}}\right).$$

Next, from standard Rademacher complexity bounds (Bartlett and Mendelson, 2002), we are guaranteed that with probability at least $1 - \delta$ we have that

$$\sup_{(\boldsymbol{\alpha}, \boldsymbol{\beta}) \in \Delta_{L'} \times \Delta_L} \left| \frac{1}{N} \sum_{i=1}^{N} f_{\boldsymbol{\alpha}, \boldsymbol{\beta}}(S_i, \mathcal{Y}_i) - \mathbb{E}_{(S, \mathcal{Y}) \sim \mathcal{D}}\big[f_{\boldsymbol{\alpha}, \boldsymbol{\beta}}(S, \mathcal{Y})\big] \right| = O\left(\hat{R}(\mathcal{F}, S) + \sqrt{\frac{\log(1/\delta)}{N}}\right)$$

$$= O\left(\sqrt{\frac{(L' + L)^2 L' \log(Nn)}{N}} + \sqrt{\frac{\log(1/\delta)}{N}}\right)$$

$$= O\left(\sqrt{\frac{(L' + L)^2 L' \log(Nn) + \log(1/\delta)}{N}}\right),$$

where the last equality follows from the fact that for any $x, y \geq 0$ we have $\sqrt{x} + \sqrt{y} \leq \sqrt{2(x + y)}$. Since the right hand side goes to zero as $N$ grows, it remains to choose $N$ sufficiently large that the right hand side is bounded by $\epsilon$.

Let $A = (L' + L)^2 L'$ and $B = (L' + L)^2 L' \log(n) + \log(1/\delta)$ so that the error bound is given by $O\left(\sqrt{\frac{A \log(N) + B}{N}}\right)$.

We are guaranteed that $\sqrt{\frac{A \log(N) + B}{N}} \leq \epsilon$ whenever $N \geq \frac{1}{\epsilon^2}(A \log(N) + B)$. Using the fact that for $a \geq 1$ and $b \geq 0$ we have that $x \geq 4a \log(2a) + 2b$ implies that $x \geq a \log(x) + b$ (e.g., see Lemma A.2 in (Shalev-Shwartz and Ben-David, 2014)), it follows that when $N \geq \frac{4A}{\epsilon^2} \log \frac{2A}{\epsilon^2} + \frac{2B}{\epsilon^2} = O\big(\frac{1}{\epsilon^2}(A \log \frac{A}{\epsilon^2} + B)\big)$, we also have $\sqrt{\frac{A \log(N) + B}{N}} \leq \epsilon$. Substituting $A$ and $B$ and simplifying the expression completes the proof. $\qquad\square$

# B    APPENDIX FOR EFFICIENT ALGORITHM SELECTION

## B.1    LEARNING THE MERGE FUNCTION

In this section we provide details for learning the best combination of two merge functions. We also give detailed pseudocode for our sweepline algorithm for finding the children of a node in the execution tree (see Algorithm 2) and for the complete algorithm (see Algorithm 3).

---

**Algorithm 2** Find all merges for $\mathcal{A}_{\mathrm{merge}}(\mathrm{D}_0, \mathrm{D}_1)$

---

**Input:** Set of clusters $C_1, \ldots, C_m$, merge functions $\mathrm{D}_0, \mathrm{D}_1$, parameter interval $[\alpha_{\mathrm{lo}}, \alpha_{\mathrm{hi}})$.
1. Let $\mathcal{M} = \emptyset$ be the initially empty set of possible merges.
2. Let $\mathcal{I} = \emptyset$ be the initially empty set of parameter intervals.
3. Let $\alpha = \alpha_{\mathrm{lo}}$.
4. While $\alpha < \alpha_{\mathrm{hi}}$:
   (a) Let $C_i, C_j$ be the pair of clusters minimizing $(1 - \alpha) \cdot \mathrm{D}_0(C_i, C_j) + \alpha \cdot \mathrm{D}_1(C_i, C_j)$.
   (b) For each $k, l \in [m]$, let $c_{kl} = \Delta_0/(\Delta_0 - \Delta_1)$, where $\Delta_p = \mathrm{D}_p(C_i, C_j) - \mathrm{D}_p(C_k, C_l)$ for $p \in \{0, 1\}$.
   (c) Let $c = \min\big(\{c_{kl} \mid c_{kl} > \alpha\} \cup \{\alpha_{\mathrm{hi}}\}\big)$.
   (d) Add merge $(C_i, C_j)$ to $\mathcal{M}$ and $[\alpha, c)$ to $\mathcal{I}$.
   (e) Set $\alpha = c$.
5. Return $\mathcal{M}$ and $\mathcal{I}$.

---

**Lemma 3.** *For any merge functions $\mathrm{D}_0$ and $\mathrm{D}_1$ and any clustering instance $S$, the execution tree for $\mathcal{A}_{merge}(\mathrm{D}_0, \mathrm{D}_1)$ when run on $S$ is well defined. That is, there exists a partition tree s.t. for any node $v$ at depth $t$, the same sequence of first $t$ merges is performed by $A_\alpha^{merge}$ for all $\alpha$ in node $v$'s interval.*

---
**Algorithm 3** Depth-first Enumeration of $\alpha$-linkage Execution Tree
---
**Input:** Point set $x_1, \ldots, x_n$, cluster distance functions $d_1$ and $d_2$.

1. Let $r$ be the root node of the execution tree with $r.\mathcal{N} = \{(x_1), \ldots, (x_n)\}$ and $r.I = [0,1]$.
2. Let $s$ be a stack of execution tree nodes, initially containing the root $r$.
3. Let $\mathcal{T} = \emptyset$ be the initially empty set of possible cluster trees.
4. Let $\mathcal{I} = \emptyset$ be the initially empty set of intervals.
5. While the stack $s$ is not empty:
   (a) Pop execution tree node $e$ off stack $s$.
   (b) If $e.\mathcal{N}$ has a single cluster, add $e.\mathcal{N}$ to $\mathcal{T}$ and $e.I$ to $\mathcal{I}$.
   (c) Otherwise, for each merge $(C_i, C_j)$ and interval $I_c$ returned by Algorithm 2 run on $e.\mathcal{N}$ and $e.I$:
       i. Let $c$ be a new node with state given by $e.\mathcal{N}$ after merging $C_i$ and $C_j$ and $c.I = I_c$.
       ii. Push $c$ onto the stack $s$.
6. Return $\mathcal{T}$ and $\mathcal{I}$.
---

*Proof.* The proof is by induction on the depth $t$. The base case is for depth $t = 0$, in which case we can use a single node whose interval is $[0,1]$. Since all algorithms in the family start with an empty-sequence of merges, this satisfies the execution tree property.

Now suppose that there is a tree of depth $t$ with the execution tree property. If $t = |S| - 1$ then we are finished, since the algorithms in $\mathcal{A}_{\text{merge}}(D_0, D_1)$ make exactly $|S| - 1$ merges. Otherwise, consider any leaf node $v$ of the depth $t$ tree with parameter interval $I_v$. It is sufficient to show that we can partition $I_v$ into subintervals such that for $\alpha$ in each subinterval the next merge performed is constant. By the inductive hypothesis, we know that the first $t$ merges made by $A_\alpha^{\text{merge}}$ are the same for all $\alpha \in I_v$. After performing these merges, the algorithm will have arrived at some set of clusters $C_1, \ldots, C_m$ with $m = |S| - t$. For each pair of clusters $C_i$ and $C_j$, the distance $D_\alpha(C_i, C_j) = (1 - \alpha) D_0(C_i, C_j) + \alpha D_1(C_i, C_j)$ is a linear function of the parameter $\alpha$. Therefore, for any clusters $C_i, C_j, C_k$, and $C_l$, the algorithm will prefer to merge $C_i$ and $C_j$ over $C_j$ and $C_k$ for a (possibly empty) sub-interval of $I_v$, corresponding to the values of $\alpha \in I_v$ where $D_\alpha(C_i, C_j) < D_\alpha(C_k, C_l)$. For any fixed pair of clusters $C_i$ and $C_j$, taking the intersection of these intervals over all other pairs $C_j$ and $C_k$ guarantees that clusters $C_i$ and $C_j$ will be merged exactly for parameter values in some subinterval of $I_v$. For each merge with a non-empty parameter interval, we can introduce a child node of $v$ labeled by that parameter interval. These children partition $I_v$ into intervals where the next merge is constant, as required. $\qquad\square$

**Lemma 4.** *Let $C_1, \ldots, C_m$ be a collection of clusters, $D_0$ and $D_1$ be any pair of merge functions, and $[\alpha_{lo}, \alpha_{hi})$ be a subset of the parameter space. If there are $M$ distinct cluster pairs $C_i, C_j$ that minimize $D_\alpha(C_i, C_j)$ for values of $\alpha \in [\alpha_{lo}, \alpha_{hi})$, then the running time of Algorithm 2 is $O(Mm^2K)$, where $K$ is the cost of evaluating the merge functions $D_0$ and $D_1$.*

*Proof.* The loop in step 4 of Algorithm 2 runs once for each possible merge, giving a total of $M$ iterations. Each iteration finds the closest pair of clusters according to $D_\alpha$ using $O(m^2)$ evaluations of the merge functions $D_0$ and $D_1$. Calculating the critical parameter value $c$ involves solving $O(m^2)$ linear equations whose coefficients are determined by four evaluations of $D_0$ and $D_1$. It follows that the cost of each iteration is $O(m^2K)$, where $K$ is the cost of evaluating $D_0$ and $D_1$, and the overall running time is $O(Mm^2K)$. $\qquad\square$

**Theorem 2.** *Let $S = \{x_1, \ldots, x_n\}$ be a clustering instance and $D_0$ and $D_1$ be any two merge functions. Suppose that the execution tree of $\mathcal{A}_{merge}(D_0, D_1)$ on $S$ has $E$ edges. Then the total running time of Algorithm 3 is $O(En^2K)$, where $K$ is the cost of evaluating $D_0$ and $D_1$ once.*

*Proof.* Fix any node $v$ in the execution tree with $m$ clusters $C_1, \ldots, C_m$ and $M$ outgoing edges (i.e., $M$ possible merges from the state represented by $v$). We run Algorithm 2 to determine the children of $v$, which by Lemma 4 costs $O(Mn^2K)$, since $m \leq n$. Summing over all non-leaves of the execution tree, the total cost is $O(En^2K)$. In addition to computing the children of a given node, we need to construct the children nodes, but this takes constant time per child. $\qquad\square$

In this section we provide details for learning the best combination of two metrics. We also give detailed pseudocode for our sweepline algorithm for finding the children of a node in the execution tree (see Algorithm 4) and for the complete algorithm (see Algorithm 5).

**Lemma 5.** *For any metrics* $d_0$ *and* $d_1$ *and any clustering instance* $S$, *the execution tree for the family* $\mathcal{A}_{metric}(d_0, d_1)$ *when run on* $S$ *is well defined. That is, there exists a partition tree s.t. for any node* $v$ *at depth* $t$, *the same sequence of first* $t$ *merges is performed by* $A_\beta^{metric}$ *for all* $\beta$ *in node* $v$*'s interval.*

*Proof.* The proof is by induction on the depth $t$. The base case is for depth $t = 0$, in which case we can use a single node whose interval is $[0, 1]$. Since all algorithms in the family start with an empty-sequence of merges, this satisfies the execution tree property.

Now suppose that there is a tree of depth $t$ with the execution tree property. If $t = |S| - 1$ then we are finished, since the algorithms in $\mathcal{A}_{metric}(d_0, d_1)$ make exactly $|S| - 1$ merges. Otherwise, consider any leaf node $v$ of the depth $t$ tree with parameter interval $I_v$. It is sufficient to show that we can partition $I_v$ into subintervals such that for $\beta$ in each subinterval the next merge performed is constant. By the inductive hypothesis, we know that the first $t$ merges made by $A_\beta^{metric}$ are the same for all $\beta \in I_v$. After performing these merges, the algorithm will have arrived at some set of clusters $C_1, \ldots, C_m$ with $m = |S| - t$. Recall that algorithms in the family $\mathcal{A}_{metric}(d_0, d_1)$ run complete linkage using the metric $d_\beta$. Complete linkage can be implemented in such a way that it only makes comparisons between pairwise point distances (i.e., is $d_\beta(x, x')$ larger or smaller than $d_\beta(y, y')$?). To see this, for any pair of clusters, we can find the farthest pair of points between them using only distance comparisons. And, once we have the farthest pair of points between all pairs of clusters, we can find the pair of clusters to merge by again making only pairwise comparisons. It follows that if two parameters $\beta$ and $\beta'$ have the same outcome for all pairwise distance comparisons, then the next merge to be performed must be the same. We use this observation to partition the interval $I_v$ into subintervals where the next merge is constant. For any pair of points $x, x' \in S$, the distance $d_\beta(x, x') = (1 - \beta) d_0(x, x') + \beta d_1(x, x')$ is a linear function of the parameter $\beta$. Therefore, for any points $x, x', y, y' \in S$, there is at most one critical parameter value where the relative order of $d_\beta(x, x')$ and $d_\beta(y, y')$ changes. Between these $O(|S|^4)$ critical parameter values, the ordering on all pairwise merges is constant, and the next merge performed by the algorithm will also be constant. Therefore, there must exist a partitioning of $I_v$ into at most $O(|S|^4)$ sub-intervals such that the next merge is constant on each interval. We let the children of $v$ correspond to the coarsest such partition.   □

**Lemma 6.** *Let* $C_1, \ldots, C_m$ *be a collection of clusters,* $d_0$ *and* $d_1$ *be any pair of metrics, and* $[\beta_{lo}, \beta_{hi})$ *be a subset of the parameter space. If there are* $M$ *distinct cluster pairs* $C_i, C_j$ *that complete linkage would merge when using the metric* $d_\beta$ *for* $\beta \in [\beta_{lo}, \beta_{hi})$, *the running time of Algorithm 4 is* $O(Mn^2)$.

*Proof.* The loop in step 4 of Algorithm 4 runs once for each possible merge, giving a total of $M$ iterations. Each iteration finds the merge performed by complete linkage using the $d_\beta$ metric, which takes $O(n^2)$ time, and then solves $O(n^2)$ linear equations to determine the largest value of $\beta'$ such that the same merge is performed. It follows that the cost of each iteration is $O(n^2)$, leading to an overall running time of $O(Mn^2)$. Note, we assume that the pairwise distances $d_\beta(x, x')$ can be evaluated in constant time. This can always be achieved by precomputing two $n \times n$ distance matrices for the base metrics $d_0$ and $d_1$, respectively.   □

**Theorem 3.** *Let* $S = \{x_1, \ldots, x_n\}$ *be a clustering instance and* $d_0$ *and* $d_1$ *be any two merge functions. Suppose that the execution tree of* $\mathcal{A}_{metric}(d_0, d_1)$ *on* $S$ *has* $E$ *edges. Then the total running time of Algorithm 5 is* $O(En^2)$.

*Proof.* Fix any node $v$ in the execution tree with $m$ clusters $C_1, \ldots, C_m$ and $M$ outgoing edges (i.e., $M$ possible merges from the state represented by $v$). We run Algorithm 4 to determine the children of $v$, which by Lemma 6 costs $O(Mn^2)$. Summing over all non-leaves of the execution tree, the total cost is $O(En^2)$.   □

## C   APPENDIX FOR EXPERIMENTS

**Clustering distributions.**

---

**Algorithm 4** Find all merges for $\mathcal{A}_{\text{metric}}(d_0, d_1)$

---

**Input:** Set of clusters $C_1, \ldots, C_m$, metrics $d_0, d_1$, parameter interval $[\beta_{\text{lo}}, \beta_{\text{hi}})$.
1. Let $\mathcal{M} = \emptyset$ be the initially empty set of possible merges.
2. Let $\mathcal{I} = \emptyset$ be the initially empty set of parameter intervals.
3. Let $\beta = \beta_{\text{lo}}$.
4. While $\beta < \beta_{\text{hi}}$:
    (a) Let $C_i, C_j$ be the pair of clusters minimizing $\max_{a \in A, b \in B} d_\beta(a, b)$.
    (b) Let $x \in C_i$ and $x' \in C_j$ be the farthest points between $C_i$ and $C_j$.
    (c) For all pairs of points $y$ and $y'$ belonging to different clusters, let $c_{yy'} = \Delta_0/(\Delta_0 - \Delta_1)$ where $\Delta_p = d_p(y, y') - d_p(x, x')$ for $p \in \{0, 1\}$.
    (d) Let $c = \min\big(\{c_{yy'} \mid c_{yy'} > \beta\} \cup \{\beta_{\text{hi}}\}\big)$.
    (e) Add merge $(C_i, C_j)$ to $\mathcal{M}$ and $[\beta, c)$ to $\mathcal{I}$.
    (f) Set $\beta = c$.
5. Return $\mathcal{M}$ and $\mathcal{I}$.

---

**Algorithm 5** Depth-first Enumeration of $\beta$-linkage Execution Tree

---

**Input:** Point set $x_1, \ldots, x_n$, cluster distance functions $d_1$ and $d_2$.
1. Let $r$ be the root node of the execution tree with $r.\mathcal{N} = \{(x_1), \ldots, (x_n)\}$ and $r.I = [0, 1]$.
2. Let $s$ be a stack of execution tree nodes, initially containing the root $r$.
3. Let $\mathcal{T} = \emptyset$ be the initially empty set of possible cluster trees.
4. Let $\mathcal{I} = \emptyset$ be the initially empty set of intervals.
5. While the stack $s$ is not empty:
    (a) Pop execution tree node $e$ off stack $s$.
    (b) If $e.\mathcal{N}$ has a single cluster, add $e.\mathcal{N}$ to $\mathcal{T}$ and $e.I$ to $\mathcal{I}$.
    (c) Otherwise, for each merge $(C_i, C_j)$ and interval $I_c$ returned by Algorithm 4 run on $e.\mathcal{N}$ and $e.I$:
        i. Let $c$ be a new node with state given by $e.\mathcal{N}$ after merging $C_i$ and $C_j$ and $c.I = I_c$.
        ii. Push $c$ onto the stack $s$.
6. Return $\mathcal{T}$ and $\mathcal{I}$.

---

*MNIST Subsets.* Our first distribution over clustering tasks corresponds to clustering subsets of the MNIST dataset (LeCun et al., 1998), which contains 80,000 hand-written examples of the digits $0$ through $9$. We generate a random clustering instance from the MNIST data as follows: first, we select $k = 5$ digits from $\{0, \ldots, 9\}$ at random, then we randomly select 200 examples belonging to each of the selected digits, giving a total of $n = 1000$ images. The target clustering for this instance is given by the ground-truth digit labels. We measure distances between any pair of digits in terms of the the Euclidean distance between their images represented as vectors of pixel intensities.

*CIFAR-10 Subsets.* We also consider a distribution over clustering tasks that corresponds to clustering subsets of the CIFAR-10 dataset (Krizhevsky, 2009). This dataset contains 6000 images of each of the following classes: airplane, automobile, bird, cat, deer, dog, frog, horse, ship, and truck. Each example is a $32 \times 32$ color image with 3 color channels. We pre-process the data to obtain neural-network feature representations for each example. We include 50 randomly rotated and cropped versions of each example and obtain feature representations from layer 'in4d' of a pre-trained Google inception network. This gives a $144$-dimensional feature representation for each of the $3000000$ examples (50 randomly rotated copies of the 6000 examples for each of the 10 classes). We generate clustering tasks from CIFAR-10 as follows: first, select $k = 5$ classes at random, then choose $50$ examples belonging to each of the selected classes, giving a total of $n = 250$ images. The target clustering for this instance is given by the ground-truth class labels. We measure distance between any pair of images as the distance between their feature embeddings.

*Omniglot Subsets.* Next, we consider a distribution over clustering tasks corresponding to clustering subsets of the Omniglot dataset (Lake et al., 2015). The Omniglot dataset consists of written characters from 50 different alphabets with a total of 1623 different characters. The dataset includes 20 examples of each character, leading to a total of 32,460 examples. We generate a random clustering instance from the Omniglot data as follows: first, we choose one of the alphabets at random. Next, we choose $k$ uniformly in $\{5, \ldots, 10\}$ and choose $k$ random characters from that alphabet. The clustering instance includes $20k$ examples and the target clustering is given by the ground-truth character labels.

We use two different distance metrics on the Omniglot dataset. First, we use the cosine distance between neural network feature embeddings. The neural network was trained to perform digit classification on MNIST. Second, each example has both an image of the written character, as well as the stroke trajectory (i.e., a time series of $(x, y)$ coordinates of the tip of the pen when the character was written). We also use the following distance defined in terms of the strokes: Given two trajectories $s = (x_t, y_t)_{t=1}^T$ and $s' = (x'_t, y'_t)_{t=1}^T$, we define the distance between them by $d(s, s') = \frac{1}{T+T'} \left( \sum_{t=1}^T d\big((x_t, y_t), s'\big) + \sum_{t=1}^{T'} d\big((x'_t, y'_t), s\big) \right)$, where $d\big((x_t, y_t), s'\big)$ denotes the Euclidean distance from the point $(x_t, y_t)$ to the closest point in $s'$. This is the average distance from any point from either trajectory to the nearest point on the other trajectory. This hand-designed metric provides a complementary notion of distance to the neural network feature embeddings.

*Places2 Subsets.* The Places2 dataset consists of images of 365 different place categories, including "volcano", "gift shop", and "farm" (Zhou et al., 2017). To generate a clustering instance from the places data, we choose $k$ randomly from $\{5, \ldots, 10\}$, choose $k$ random place categories, and then select 20 random examples from each chosen category. We restrict ourselves to the first 1000 images from each class.

We use two metrics for this data distribution. First, we use cosine distances between feature embeddings generated by a VGG16 network pre-trained on imagenet. In particular, we use the activations just before the fully connected layers, but after the max-pooling is performed, so that we have $512$-dimensional feature vectors. Second, we compute color histograms in HSV space for each image and use the cosine distance between the histograms. In more detail, we partition the hue space into $8$ bins, the saturation space into $2$ bins, and the value space into $4$ bins, resulting in a 64-dimensional histogram counting how frequently each quantized color appears in the image. Two images are close under this metric if they contain similar colors.

*Places2 Diverse Subsets.* We also construct an instance distribution from a subset of the Places2 classes which have diverse color histograms. We expect the color histogram metric to perform better on this distribution. To generate a clustering instance, we pick $k = 4$ classes from aquarium, discotheque, highway, iceberg, kitchen, lawn, stage-indoor, underwater ocean deep, volcano, and water tower. We include $50$ randomly sampled images from each chosen class, leading to a total of $n = 200$ points per instance.

*Synthetic Rings and Disks.* We consider a two dimensional synthetic distribution where each clustering instance has 4 clusters, where two are ring-shaped and two are disk-shaped. To generate each instance we sample 100 points uniformly at random from each ring or disk. The two rings have radiuses $0.4$ and $0.8$, respectively, and are both centered at the

origin. The two disks have radius $0.4$ and are centered at $(1.5, 0.4)$ and $(1.5, -0.4)$, respectively. For this data, we measure distances between points in terms of the Euclidean distance between them.

**Average Number of Discontinuities**. Next we report the average number of discontinuities in the loss function for a clustering instance sampled from each of the distributions described above for each of the learning tasks we consider. In all cases, the average number of discontinuities is many orders of magnitude smaller than the upper bounds. The metric learning problems tend to have more discontinuities than learning the best merge function. Surprisingly, even though our only worst-case bound on the number of discontinuities when interpolating between average and complete linkage is exponential in $n$, the empirical number of discontinuities is always smaller than for interpolating between single and complete linkage. The results are shown in Table 1.

| Distribution | Task | max $n$ | Average # Discontinuities |
|---|---|---|---|
| Omniglot | SC | 200 | 59.4 |
| Omniglot | AC | 200 | 33.9 |
| Omniglot | metric | 200 | 201.1 |
| MNIST | SC | 1000 | 362.6 |
| MNIST | AC | 1000 | 282.0 |
| Rings and Disks | SC | 400 | 29.0 |
| Rings and Disks | AC | 400 | 18.3 |
| CIFAR-10 | SC | 250 | 103.2 |
| CIFAR-10 | AC | 250 | 66.2 |
| Places2 | metric | 200 | 241.0 |
| Places2 Diverse | metric | 200 | 269.6 |

Table 1: Table of average number of discontinuities for a piecewise constant loss function sampled from each distribution and learning task. Task 'SC' corresponds to interpolating between single and complete linkage, 'AC' is interpolating between average and complete linkage, and 'metric' is interpolating between two base metrics. The column labeled "max $n$" is an upper bound on the size of each clustering instance.

