# OpenReview forum: "Learning to Link"
_ICLR.cc/2020/Conference — Accept (Poster)_

### Official Review · AnonReviewer2 · 2019-10-22
**Official Blind Review #2**

**Rating:** 6

**Review:**

In this paper, the authors propose an approach to learning combinations of (instance-wise) distance metrics and (cluster-wise) merge functions to optimally cluster instances from a  particular data distribution. In particular, given a set of clustering instances (each of which is a set of instances from the domain and their cluster assignment), a set of distance metrics, and a set of merge functions, the proposed approach aims to learn a convex combination of the distance metrics and merge functions to reconstruct the given clusterings.

The paper has two main contributions. First, a PAC learning type of guarantee is given on the quality of the learned clustering approach. Second, an efficient data structure for identifying the convex combinations is given. A small set of experiments suggests that, in practice, the learned combinations can outperform using single distance metrics and merge functions.

Comments

I am not an expert in this area; I had trouble following the details of the theoretical developments. However, I appreciated that intuition was given on both what the theorems and lemmas were showing as well as the main steps of the proofs.

Concerning Theorem 1, it is not exactly clear to me what the contribution is on top of [Balcan et al., 2019]. The text mentions that they already give sample complexity guarantees in what seems like the same setting (piecewise-structured cost function).

The authors point out that depth-first traversal is a good choice here due to its memory efficiency. However, in cases where the search space is a graph rather than a tree (i.e., there are multiple paths to some nodes), then DFS can exponentially increase the work compared to breadth-first or other search strategies (e.g., [Edelkamp and Schroedl, 2012]). While the name suggests that the “execution tree” is, indeed, a tree, is this guaranteed to be the case? or could multiple paths lead to the same partition?

For the experimental evaluation, it seems as though there is no “test” set of clustering instances. It would be helpful to also include performance of the learned combinations on some test clustering instances to give an idea of how generalizable to approach is to other instances within the data distribution. (Of course, the main contributions of this work are the theoretical developments, so just one or two examples would be sufficient.)

For motivation, it would be helpful to give some examples where the prerequisites of this work are actually met; that is, cases where sufficiently large number of labeled cluster instances are available, but the generative mechanism of the clusters is not.

For context, it could be helpful to briefly mention how, if at all, the current results apply to widely-used clustering algorithms such as k-means or Gaussian mixture models.

Typos, etc.

The references are somewhat inconsistently formatted. Also, some proper nouns in titles are not capitalized (e.g., “lloyd’s families”).

“leaves correspond to” -> “leaves corresponding to”

What does the “big-Oh tilde” notation in Theorem 1 mean?


**Experience Assessment:**

I do not know much about this area.

**Review Assessment: Checking Correctness Of Derivations And Theory:**

I assessed the sensibility of the derivations and theory.

**Review Assessment: Checking Correctness Of Experiments:**

I carefully checked the experiments.

**Review Assessment: Thoroughness In Paper Reading:**

I read the paper at least twice and used my best judgement in assessing the paper.

---

> ### Author Response · Authors · 2019-11-15
> **Response to Review #2**
>
> Thank you for your careful review and thoughtful comments.
>
> - The key insight behind our sample complexity guarantee is that for the proposed family of algorithms and any clustering instance, we can partition the parameter space into a small number of simple regions where the algorithm output is constant. We rely on the results of Balcan et al. (2019) only for the last step in the proof to convert this structural property into a sample complexity guarantee. In the camera ready version we will clarify this and provide a direct proof of our sample complexity bound - the proof for our specific setting does not really need the subtle machinery of  that work.
>
> - The algorithms we propose in Section 3 completely enumerate the leaves of the execution tree (i.e., they don't stop early once they've found a promising leaf), and therefore the number of nodes visited by DFS and BFS are identical. The key difference is that DFS only keeps one path from root to leaf in memory at a time, and for this algorithm family we are guaranteed that the depth of the tree is $|S|-1$. On the other hand, BFS keeps one level of the tree in memory at a time, which can be significantly larger (e.g., for interpolating between single and complete linkage, our best bound on the width of a level is $O(|S|^8)$. We will clarify this in the camera ready version of the paper.
>
> - We briefly describe a few examples of where the prerequisites of our work are met in the introduction. In general, we have in mind any application where we face a sequence of clustering tasks and we can ask a human to provide the target clusterings for those tasks (e.g., clustering the news articles that appear day to day by topic). In these situations, having a low sample complexity is crucial because we do not want to ask for many target clusterings. We will expand and emphasize the examples in the camera ready version of the paper.
>
> - While our results do not apply to the $k$-means algorithm or Gaussian mixture models, we do not view this as a shortcoming given that hierarchical clustering is widely used and a classic research area. Our related work discusses relationships between our work and a related paper for learning initialization procedures for Lloyd's method for $k$-means clustering. We will include further discussion of these relationships and the applicability of our results.
>
> - We agree that it would be interesting to validate our sample complexity results empirically. See our response to R3's similar comment.
>
> - We will correct the typos and ambiguities you found, thanks!

---

> > ### Comment · AnonReviewer2 · 2019-11-15
> > **RE: Author response**
> >
> > I have read the other reviews and authors' responses. They do not change my view of the paper ("weak accept"). That is, I appreciate the theoretical contribution (from a non-expert perspective), but like the other reviewers, I believe stronger empirical results would really strengthen the arguments presented in this work.

---

### Official Review · AnonReviewer3 · 2019-10-25
**Official Blind Review #3**

**Rating:** 6

**Review:**

Summary:

This paper proposed a data-driven method of selecting a linkage-based clustering algorithm from a large space. The space of algorithms is parameterized by two sets of parameters which indicate the convex combinations of metrics and merge functions. They analyze the sample complexity for small generalization error. An efficient algorithm for searching an empirically optimal algorithm is proposed.

Comments:

In general, I think this is a good quality paper.
- Selecting a clustering algorithm from a large space by a data-driven method is an interesting and sound idea, which makes a lot of sense to me.
- The theorem for generalization error is strong.

It can be further improved in the following aspects (mainly the experiments).
- The curves in Fig 3 all look smooth, so I wondered whether one can simply apply a grid search on [0.1,0.2,...,1.0], the obtained algorithm should also be very good. To demonstrate the advantage and necessity of the proposed search algorithm, I think it better to either conduct an experiment with a higher dimensional search space (instead of only searching \alpha) or demonstrate a case when there is a sharp turn near the optimal point, so that grid search won't work well.
- Although the authors have proved the generalization error, it is still better to empirically validate the theoretical result, by showing the training and testing errors along with varying sample sizes.

Overall I like the idea and the theoretical analysis in this paper, but the experimental results could be further improved. Therefore I vote for weak acceptance.


----- after reading the response --

I'd like to thank the authors for giving more explanations. Theoretically, I understand the advantages of the proposed algorithm, but still, it is more convincing if stronger experiments can be conducted.

My score does not change, but overall I advocate to accept this paper.


**Experience Assessment:**

I have read many papers in this area.

**Review Assessment: Checking Correctness Of Derivations And Theory:**

I assessed the sensibility of the derivations and theory.

**Review Assessment: Checking Correctness Of Experiments:**

I assessed the sensibility of the experiments.

**Review Assessment: Thoroughness In Paper Reading:**

I read the paper at least twice and used my best judgement in assessing the paper.

---

> ### Author Response · Authors · 2019-11-15
> **Response to Review #3**
>
> Thank you for your careful review and thoughtful comments.
>
> - Yes, for the specific distributions in our experiments section, using grid search with a sufficiently fine grid would find nearly optimal parameters. However, there are clustering distributions for which the expected cost is not smooth and the set of approximately optimal parameters is an arbitrarily small interval (we will include such an example in the camera ready version). Our proposed methods have two significant advantages over grid search. First, they are guaranteed to return empirically optimal parameters even when the performance is not smooth. Second, they are more efficient than running grid search with a very fine grid. To see this, observe that if the grid has $G$ points, we must run the clustering algorithm $G$ times on every training clustering instance. In contrast, the cost of our proposed methods scales with the number of discontinuities for each instance. If $G$ is bigger than the average number of discontinuities, then the grid search is actually more computationally expensive.
>
> - We agree that it would be interesting to validate our theoretical claims empirically by showing that the performance of parameters are similar across training and testing datasets of various sizes. Our current experimental results support our theory by showing that for several natural distributions over clustering tasks, we can obtain large improvements in performance by combining a pair of merge functions or a pair of metrics. They also show that our proposed optimization algorithms are efficient enough to run on realistically sized clustering instances.

---

### Official Review · AnonReviewer1 · 2019-10-30
**Official Blind Review #1**

**Rating:** 6

**Review:**

This paper studies the problem of learning both the distance metric and a linkage rule from clustering examples. Suppose we have L metrics d_1, …, d_L and L’ linkage rules for hierarchical agglomerative clustering, D_1, …, D_L’ where each rule is a 2-point-based merge function (i.e. computes the distance between some two points in the clusters, examples of such functions are single-linkage and complete-linkage). The paper considers the problem of finding the convex combination of the distance functions and linkage rules which best fits the data. The main result (Theorem 1) is an \tildeO((L’ + L)^2 L’ /eps^2) uniform convergence bound on the number of clustering instances which are required to learn up to expected loss \eps the best possible convex combination. The key technical part of the proof is showing that for any fixed clustering the loss function is piecewise-constant with a small number of simple pieces. The overall approach is based on Balcan et al.’17 who solve the case when the distance metric is known but the linkage rule is to be learned and Balcan et al. ‘19 who give techniques for the piecewise constant case. Some further results are given which are specific to learning a mix of two merge functions under a single distance metric and the best combination of two metrics when using the complete linkage merge function. Experimental results are given on MNIST, CIFAR-10 and some other fairly small datasets.

The paper makes a somewhat interesting contribution to the area, but I think can only be seen as a basic step in the general direction. Most of the interesting merge functions used for HAC don’t boil down to simple 2-point-merge rules (average-linkage, Ward’s method, etc.). The sample complexity of the problem is rather prohibitive. In particular, it is unclear to me why the experimental setup in the paper is consistent with the theoretical model -- when i.i.d. clusterings should be sampled from a distribution, why is it ok to just sample 5 random classes from MNIST a bunch of times? In this case the ground truth clustering is fixed and you sample some subset of classes from it each time. This seems like a much simpler setup compared to the general setting considered in the paper.  I would expect a real experimental setup to have all n points be fixed, then you have a distribution over different clusterings on the same set of points which you sample from each time.


**Experience Assessment:**

I have published in this field for several years.

**Review Assessment: Checking Correctness Of Derivations And Theory:**

I assessed the sensibility of the derivations and theory.

**Review Assessment: Checking Correctness Of Experiments:**

I assessed the sensibility of the experiments.

**Review Assessment: Thoroughness In Paper Reading:**

I read the paper thoroughly.

---

> ### Author Response · Authors · 2019-11-15
> **Response to Review #1**
>
> Thank you for your careful review and thoughtful comments.
>
> - At the end of Section 2 we discuss how our analysis can be extended to include merge functions beyond 2-point-based merges. In the camera ready version of the paper we will include clarifications and additional details. We only use the 2-point-based property in Lemma 2. First, we show that restricted to beta belonging to any region of the partition constructed in Lemma 1, any 2-point-based merge function is a linear function of the metric parameter $\beta$. Second, we use it to count the total number of quadratic functions that must be included in the set Q. We can extend the result to also hold when one of the merge functions is chosen to be average-linkage using the following insights: first, the average-linkage distance between a pair of clusters is always a linear function of the metric parameter $\beta$. Second, we can replace the bound of $O(|S|^{4L'})$ on the size of $\mathcal{Q}$ by $O(3^{|S|})$ since that is a bound on the number of ways to choose two clusters from $|S|$ points. This exponential increase in the size of $\mathcal{Q}$ corresponds to a linear dependence on $|S|$ in our final sample complexity, since we depend only on the log size of $\mathcal{Q}$.
>
> - The motivating setting for our experiments is described in the introduction. We are thinking about situations where we encounter a collection of related clustering tasks and where the target clustering is consistent from task to task (e.g., each day we cluster the articles appearing in a newspaper and our goal is always to cluster them by topic). The distributions in our experiments model this type of situation. Each instance includes a random set of points (drawn from a larger classification dataset) and our goal is to find an algorithm that best recovers the target clustering given by the ground-truth labels. If we think of the points as being news articles and the class labels being the unknown topics, then this fits well with our formal problem setup and is consistent with our theoretical results. In contrast, if we were to keep the points fixed but vary the target clustering, no single algorithm will have good performance, since each algorithm can produce only one clustering for the points.

---

### Decision · Program_Chairs · 2019-12-19

**Decision:**

Accept (Poster)

**Comment:**

All reviewers come to agreement that this is a solid paper worth publishing at ICLR; the authors are encouraged to incorporate additional comments suggested by reviewers.